# THICKER AND QUICKER: A JUMBO TOKEN FOR FAST PLAIN VISION TRANSFORMERS

**Anthony Fuller**[1,3]**, Yousef Yassin**[1]**, Daniel G. Kyrollos**[1]**, Evan Shelhamer**[1,2,3]★**, James R. Green**[1]★

Carleton University[1], University of British Columbia[2], Vector Institute[3]
★ equal advising

## ABSTRACT

ViTs are general and accurate, and address many tasks, but ViTs are slow, and are not always practical when efficiency is key. Existing methods for faster ViTs design hybrid non-ViT architectures, losing generality, or shrink their tokens, sacrificing accuracy. Many non-ViT architectures are both fast and accurate. Yet, without significant modifications, they cannot do what ViTs can: process other input shapes, pre-train by SOTA self-supervised learning, reduce computation by dropping tokens, and more. We make ViTs faster by reducing patch token width while *increasing* global token width by adding a new Jumbo token. Our wider Jumbo token is processed by its own wider FFN to increase model capacity. Yet our Jumbo FFN is efficient: it processes a single token, for speed, and its parameters are shared across all layers, for memory. Crucially, our Jumbo is *attention-only* and *non-hierarchical*, like a plain ViT, so it is simple, scalable, flexible, and compatible with ViT methods new and old. Jumbo improves over ViT baselines with Registers from Nano to Large scales *while maintaining speed/throughput* on ImageNet-1K ($\uparrow 0.1-13\%$). Jumbo also improves segmentation ($\uparrow 1.9-3.1\%$ on ADE20K), MAE pre-training ($\uparrow 4.9\%$ linear probing on ImageNet-1K), test-time adaptation ($\uparrow 5.2\%$ on ImageNet-C), and time series modeling. Our Jumbo models even achieve better speed-accuracy trade-offs than *specialized non-ViT* compute-efficient models, while maintaining plain-ViT compatibility for practicality. Code and weights are available: `https://github.com/antofuller/jumbo`

## 1 INTRODUCTION: ARCHITECTURE, ACCURACY, AND EFFICIENCY

For most model sizes, the vision transformer (ViT; Dosovitskiy et al. (2021)) is the go-to architecture in computer vision—powering foundation models like DINOv2 (Oquab et al., 2024), language-aligned models like CLIP (Radford et al., 2021a), segmentation models like SAM (Kirillov et al., 2023), 3D vision models like DUST3R (Wang et al., 2024), and diffusion models like DiT (Peebles & Xie, 2022). These are all "plain" ViTs, which are crucially *attention-only* and *non-hierarchical*.

At the smallest scales—offering the *highest speeds/throughputs*—plain ViTs are not competitive with highly specialized architectures (Yun & Ro, 2024). We attribute the worse accuracy-speed of plain ViTs to their *width* (number of channels). Existing work scales width *equally* across all tokens and layers so higher speed requires lower width: ViT-Base (768) $\rightarrow$ ViT-Small (384) $\rightarrow$ ViT-Tiny (192).

We scale width differently across tokens and equally across layers. Our architecture adds a **Jumbo** token, which replaces the conventional `CLS` token, that is $J\times$ wider than the patch tokens, with its own wider feed-forward network (FFN), to effectively and efficiently boost model capacity. For self-attention, the Jumbo token is split into $J\times$ as many tokens/heads, but the Jumbo FFN is only applied to the one (merged) token to reduce time and shared across layers to reduce memory. Jumbo keeps the defining traits of a plain ViT—attention-only and non-hierarchical—so Jumbo applies anywhere a plain ViT does but at higher speed.

The simplicity of ViTs is due to their attention-only and non-hierarchical architecture. Multiple uses of ViTs rely on this architectural "interface" for their computation and function. For instance, this interface enables efficient sparse computation through masking/token dropping. Random token dropping enables efficient training (Liu et al., 2023; Dehghani et al., 2024; Leroy et al., 2024) and learned token dropping enables efficient deployment (Bolya et al., 2023; Fuller et al., 2025). Several

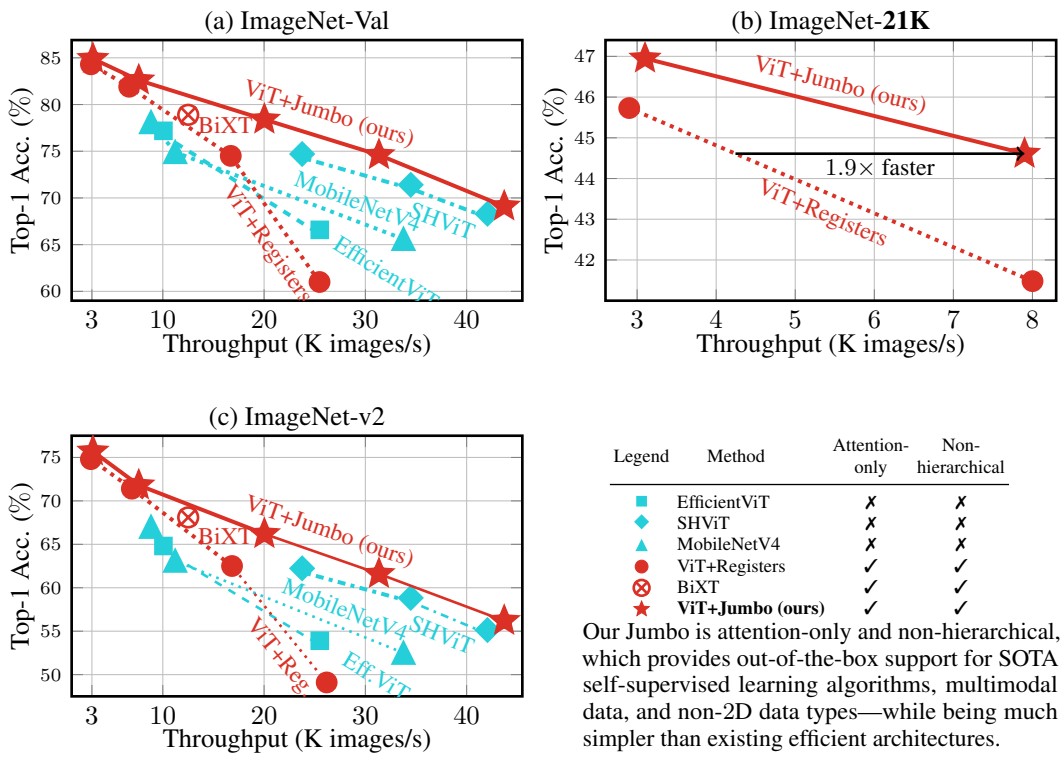

Figure 1: Plain ViTs are in red, and others are in blue. ViT+Jumbo outperforms SOTA compute-efficient architectures — *while maintaining the advantages of plain ViTs*. ViT+Jumbo outperforms ViT+Registers on ImageNet-1K and the more challenging ImageNet-21K dataset. Throughput is measured on an RTX 4090 GPU using PyTorch 2.6.0, `torch.compile`, and a 512 batch size.

SOTA self-supervised learning (SSL) algorithms require token dropping for learning (He et al., 2022; Garrido et al., 2024; Wei et al., 2025; Venkataramanan et al., 2025). This same interface enables flexible processing of different input shapes, like time series (Nie et al., 2023) or video (Arnab et al., 2021). Moreover, many extensions and applications—from object detection and segmentation heads (Fang et al., 2023; Zhang et al., 2022) to test-time adaptation algorithms (Niu et al., 2023)—are *designed for this plain ViT interface*. Architectures that maintain ViT compatibility inherit all of this.

Our experiments show that Jumbo improves speed-accuracy performance across tasks, datasets, and modalities. ❶ **Image classification**: Jumbo outperforms ViTs with Registers (Darcet et al., 2024) by $0.1-13\%$ on ImageNet-1K and $1.2-3.1\%$ on ImageNet-21K *while maintaining throughput* and achieves the pareto frontier vs. compute-efficient architectures. ❷ **Segmentation**: Jumbo gains $1.9-3.1\%$ on ADE20K (Zhou et al., 2017) using a standard segmentation head. ❸ **Self-Supervised Learning (SSL)**: Jumbo improves MAE (He et al., 2022) pre-training measured with linear probing by $4.9\%$ on ImageNet-1K at ViT-Base scale—this ViT-Base+Jumbo *ties* the ViT-Large baseline, with $2.3\times$ fewer parameters, $3.5\times$ fewer FLOPs, and $3.1\times$ higher throughput. ❹ **Test-time adaptation (TTA)**: Jumbo is more accurate and more robust with $5.2\%$ improvement on ImageNet-C using a SOTA adaptation method for transformers (SAR (Niu et al., 2023)). ❺ **Time series**: Jumbo generalizes beyond vision to rank first across 20 time series benchmarks vs. transformer baselines.

Jumbo is such an efficient ViT-compatible architecture that it outperforms *highly specialized* existing architectures on ImageNet-1K (Fig. 1). This is notable because such compute-efficient architectures (Chen et al., 2023; Howard et al., 2017) sacrifice generality and compatibility with other techniques and applications. Even efficient architectures based on ViTs include convolutions, hierarchy, and batch normalization (Yun & Ro, 2024; Vasu et al., 2023b; Cai et al., 2023) that make them **incompatible** out of the box with SSL by MAE, TTA by SAR, time series, ViT heads, etc. *Jumbo delivers compute efficiency while maintaining plain-ViT compatibility.*

## 2 BACKGROUND AND RELATED WORK: GENERALISTS AND SPECIALISTS

### 2.1 VISION TRANSFORMERS: SIMPLE, FLEXIBLE, BUT NOT YET FAST

**Jumbo extends ViTs.** A ViT splits an image into a patch grid, $\mathbb{R}^{Y \times X \times C} \to \mathbb{R}^{N_y \times N_x \times P_y \times P_x \times C}$, where $C$ is the number of channels, $Y$ / $X$ are the image height / width, $N_y$ / $N_x$ are the grid height / width, and $P_y$ / $P_x$ are the patch height / width in pixels (equal to $\frac{Y}{N_y}$ / $\frac{X}{N_x}$). Next, they flatten the grid into a sequence and flatten the patches into vectors, $\mathbb{R}^{N_y \times N_x \times P_y \times P_x \times C} \to \mathbb{R}^{N \times D_{pix}}$, where $N$ is the number of patches (equal to $N_y \cdot N_x$), and $D_{pix}$ is the number of pixel values per patch (equal to $P_y \cdot P_x \cdot C$). Next, they apply a learnable linear projection to form patch embeddings, $\mathbb{R}^{N \times D_{pix}} \to \mathbb{R}^{N \times D}$, where $D$ is the token width, also known as the embedding dimension. Next, they add position embeddings to patch embeddings. These operations produce patch tokens $\mathbf{x}^P \in \mathbb{R}^{N \times D}$ that represent local information—typically a $16 \times 16$ px square. Crucially for us, ViTs prepend a learnable CLS token $\mathbf{x}^{\text{CLS}}$ to the sequence of patch tokens, $\mathbf{x} = \mathbf{x}^{\text{CLS}} \|_0 \mathbf{x}^P \in \mathbb{R}^{(N+1) \times D}$, where $\|_0$ denotes concatenation along the $0^{\text{th}}$ (sequence) dimension. Finally, the input $\mathbf{x}$ is processed by a plain transformer and the CLS token, having attended to all other tokens, can serve as the global representation of the image. ViT sizes vary w.r.t. depth and width. ViT-Large has 24 layers, others have 12 layers, while the widths vary $\{96, 128, 192, 384, 768, 1024\}$, corresponding to names $\{$Pico, Nano, Tiny, Small, Base, Large$\}$. Narrower ViTs require less computation and are thus faster.

A standard image size of $224 \times 224$ px and a standard patch size of $16 \times 16$ px result in 196 local tokens. A *single* CLS token—designed to aggregate global information for classification—provisions $1/197^{\text{th}}$ of a model's representational capacity to global information (and this fraction decreases with larger images and/or smaller patches). This allocation is imbalanced, and may not be optimal. Recent work finds evidence to support this intuition and proposes a fix: register tokens.

**Registers.** Darcet et al. (2024) find that ViTs learn to repurpose some patch tokens to behave like additional CLS tokens by collecting global information and discarding patch-specific local information. The same work proposes a fix: prepend extra learnable tokens—called registers $\mathbf{x}^{\text{Reg}} \in \mathbb{R}^{R \times D}$, where $R$ is the number of registers—to the input sequence, $\mathbf{x} = \mathbf{x}^{\text{CLS}} \|_0 \mathbf{x}^{\text{Reg}} \|_0 \mathbf{x}^P \in \mathbb{R}^{(N+R+1) \times D}$. Registers improve accuracy (by $\sim 0.4\%$ on ImageNet-1K (Russakovsky et al., 2015) at ViT-Base) and reduce attention map artifacts/noise by provisioning more global capacity.

Registers are elegant, simple, and keep the plain ViT interface. In theory, registers can benefit any plain, non-causal transformer. These advantages account for registers' *significant and immediate* impact including in applications beyond images (Dong et al., 2024; Vaquero et al., 2024; Leigh et al., 2024; Messaoud et al., 2025; Hu et al., 2024; Thimonier et al., 2024; Omranpour et al., 2024). Our Jumbo is inspired by ViT+Registers: see Fig. 2 for their relationship and key differences.

### 2.2 COMPUTE-EFFICIENT ARCHITECTURES: FAST, BUT NOT SIMPLE NOR FLEXIBLE

The Jumbo architecture is accurate and compute-efficient, so we highlight 3 architectures and use them as baselines for high-speed ViTs. ❶ EfficientViT (Cai et al., 2023) and ❷ SHViT (Yun & Ro, 2024) improve the efficiency of ViTs by incorporating efficient attention, pooling, and convolutional layers. ❸ MobileNetV4 (Qin et al., 2025) improves the efficiency of CNNs by leveraging many strategies (and different strategies for different model sizes). These baselines represent the SOTA in computational efficiency; please refer to Appendix A.2 for descriptions of these model architectures.

Beyond these, there is a rich literature on compute-efficient vision architectures. For example, several efficient CNN-based architectures exist (Howard, 2017; Sandler et al., 2018; Howard et al., 2019; Han et al., 2020; Tan et al., 2019; Vasu et al., 2023a); however, these are surpassed by MobileNetV4 (Qin et al., 2025). Since the invention of the ViT, there have been many compute-efficient "ViTs" that incorporate efficiencies inspired by CNN-based approaches (Vasu et al., 2023b; Mehta & Rastegari, 2021; 2022; Li et al., 2023; Pan et al., 2022; Chen et al., 2022; Li et al., 2022). SHViT (Yun & Ro, 2024) has recently surpassed these architectures. Despite their impact and ingenuity, none of these hybrid architectures meets the definition of a plain ViT, which is attention-only and non-hierarchical; they thus lose many advantages of ViTs that we wish to keep. On the other hand, BiXT (Hiller et al., 2024) models are an efficient extension of the Perceiver architecture (Jaegle et al., 2021) that keeps the attention-only and non-hierarchical properties of ViTs, which are a natural comparison to Jumbo.

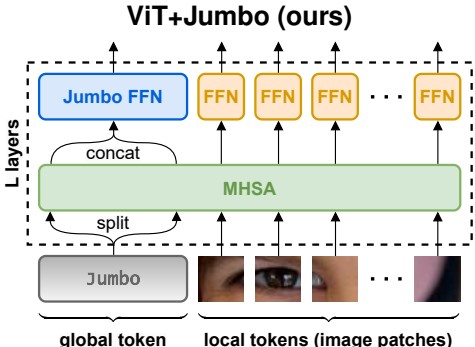
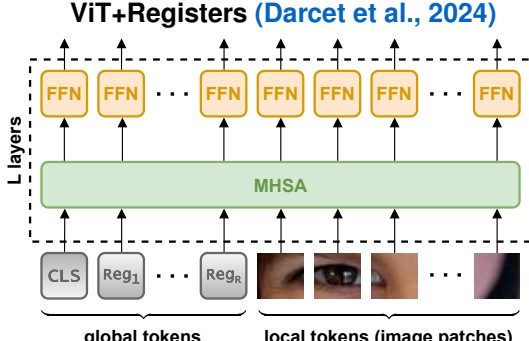

Figure 2: **(Left)** Our ViT+Jumbo method creates a wide global token that gets split into several tokens, with width equal to the patch width, prior to multi-headed self-attention (MHSA). After attention, the split Jumbo token is reassembled via concatenation, and is then processed by *its own* **FFN**. Patches are processed by **their own, shared FFN**. **(Right)** ViT+Registers creates register tokens all equal to the patch width — and all tokens are processed by **a shared FFN**. ViT+Jumbo enhances global processing as the (split) global tokens can interact via an expressive FFN, plus attention.

# 3 METHOD: A JUMBO TOKEN FOR A COMPUTE-EFFICIENT PLAIN VIT

## 3.1 DESIGN MOTIVATION AND INTUITION

**Capacity and Cost.** Although Jumbo adds a wider token and FFN, the cost is minimal. *The key insight is that a single wide token affords much greater width and more processing without slower speed.* As shown in Fig. 3, the main drivers of computational cost (FLOPs per layer) are sequence length and patch width, $D$. The FLOP contribution from our Jumbo token is comparatively negligible. Since our architecture shares Jumbo FFN parameters across all ViT layers, its memory costs are also minimal.

**Non-hierarchical and attention-only.** Jumbo preserves the non-hierarchical shape of ViTs (also known as columnar or isotropic shape). By foregoing convolutions, spatial information only moves through attention. These two properties have several advantages that we now discuss.

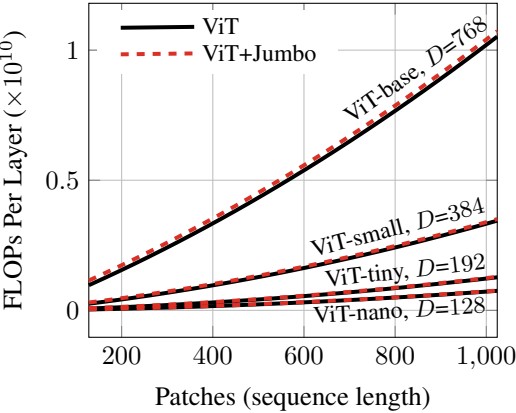

Figure 3: The cost of layers is largely determined by the number of patches and their width $D$. The cost of our Jumbo token ($J{=}6$) is negligible.

**Token Dropping / Masking.** Although convolutions are capable of processing a sparse subset of patches via sparse compute kernels, these kernels can be complex, challenging to use, and require updating when new hardware arrives. Furthermore, sparse convolutional kernels will never be as efficient as simply indexing from a sequence—i.e., how transformers drop tokens. As a comparison, ConvNeXt V2 (Woo et al., 2023) reports a $1.3\times$ speedup using a $60\%$ masking ratio with the Minkowski Engine v0.5.4 (Choy et al., 2019). Conversely, MAE (He et al., 2022) report $2.8 - 4.1\times$ speedups using a $75\%$ masking ratio with plain ViTs. *Efficient token dropping is required for SOTA SSL algorithms* (Assran et al., 2023; Fu et al., 2024; Garrido et al., 2024; Wei et al., 2025; Venkataramanan et al., 2025; Oquab et al., 2024). Token dropping also speeds up supervised training (Dehghani et al., 2024). We demonstrate Jumbo's token dropping ability in subsections 4.2 and 4.4.

**Other Data Modalities and Shapes.** These properties explain the input flexibility of transformers, which Jumbo keeps. For example, users need only adjust tokenization strategies for 1D time series, 3D point clouds, or multimodal data. We show a 1D time series application of Jumbo in subsection 4.6 and preliminary application to vision-language data in subsection 4.7.

**Plain ViT's Ecosystem.** These two properties—non-hierarchical and attention-only—maintain support for methods invented for the plain ViT. For example, segmentation and object detection heads (Fang et al., 2023; Liu et al., 2025; Zhang et al., 2022), which expect ViT's unpooled feature map; test-time adaptation methods (Niu et al., 2023), designed for the LayerNorm (Ba et al., 2016) *not* BatchNorm (Ioffe & Szegedy, 2015); and attention improvements, such as Flash Attention (Dao et al., 2022), which can speed up self-attention by $>5\times$. *Jumbo supports these innovations out of the box.*

Crucially, *none* of the compute-efficient architectures in subsection 2.2 immediately benefit from these advances, support other data modalities or token dropping, or integrate with the ViT ecosystem.

**Two hypotheses.** Jumbo asymmetrically increases the model capacity. Thus, ❶ we expect increasing gains due to Jumbo with *decreasing* patch token width. ❷ We expect increasing gains due to Jumbo with *increasing* task output dimensionality. We explore both of these hypotheses using experiments with ViTs of different widths and datasets of different complexities.

## 3.2 DESIGN SPECIFICS FOR TOKEN-WIDTH ASYMMETRY

Exactly like the original ViT, Jumbo computes patch embeddings, $\mathbf{x}^P \in \mathbb{R}^{N \times D}$. Unlike the original ViT, our method creates a Jumbo token that is $J$ times wider than the patch width $D$, $\mathbf{x}^{\text{Jumbo}} \in \mathbb{R}^{J \cdot D}$. Architecturally identical transformer layers then process these inputs.

Before self-attention, the Jumbo token is split into $J$ tokens, $|_J^1 \mathbf{x}^{\text{Jumbo}} : \mathbb{R}^{1 \times J \cdot D} \to \mathbb{R}^{J \times D}$, where $|_J^1$ denotes splitting into $J$ segments along the 1st (feature) dimension. Next, the split Jumbo token is concatenated with patch embeddings along the sequence dimension, $\mathbf{x} = \mathbf{x}^{\text{Jumbo}} \|_0 \mathbf{x}^P \in \mathbb{R}^{(N+J) \times D}$. This sequence is sent through a plain multi-headed self-attention layer. Afterward, the Jumbo token is extracted from the sequence by splitting along the sequence dimension, $|_2^0 \mathbf{x} : \mathbb{R}^{(N+J) \times D} \to (\mathbb{R}^{J \times D}, \mathbb{R}^{N \times D})$, where the first element contains the (still split) Jumbo token and the second element contains the patch representations. Finally, the Jumbo token is reassembled through concatenation along the channel dimension, $\mathbf{x}^{\text{Jumbo}} = \|_1 \mathbf{x}^{\text{Jumbo}} : \mathbb{R}^{J \times D} \to \mathbb{R}^{1 \times J \cdot D}$. These two splits and two concatenations add negligible runtime overhead.

After self-attention, the Jumbo token is processed by its own FFN that does not share parameters with the patch FFN. Fig. 2 indicates this by coloring the Jumbo and patch FFNs differently. After processing by all layers, we project the Jumbo token to $C$ class logits, $\mathbb{R}^{J \cdot D} \to \mathbb{R}^C$.

**Layer sharing.** We share our Jumbo FFN parameters across all layers to reduce memory use (through fewer model parameters). All other model parameters are not shared across layers, as usual. Sharing also acts as regularization. Empirically, we find sharing keeps (and sometimes increases) Jumbo's accuracy gains compared with *not* layer sharing—while effectively controlling memory use. Sharing the FFN layer is thus the default in our Jumbo architecture.

## 4 EXPERIMENTS: ACCURACY, COMPUTE EFFICIENCY, AND GENERALITY

For all experiments, we measure throughput on an RTX 4090 GPU using PyTorch 2.6.0, `torch.compile`, and a 512 batch size.

### 4.1 IMAGENET-1K EXPERIMENTS WITH COMPUTE-EFFICIENT BASELINES

**Setup.** We perform controlled experiments to evaluate Jumbo. Specifically, we train models from scratch on ImageNet-1K (Russakovsky et al., 2015) at $128 \times 128$ px for 400 epochs, then for 20 epochs at $224 \times 224$ px. We leverage distillation to improve convergence, which is a common strategy.

We train each model architecture twice, once for each learning rate {1e-3, 3e-3} (Touvron et al., 2022; Yun & Ro, 2024) using a 1024 batch size with the AdamW optimizer (Loshchilov, 2017). We report the results of the best learning rate for each model architecture. Please see Appendix A.4.1 & A.5 for hyperparameters and complete results, respectively.

**Baselines.** We choose the high-speed models for each family: ❶ ViT+Registers {Nano, Tiny, Small, Base} (Darcet et al., 2024), ❷ BiXT has 1 size (Tiny), ❸ EfficientViT {B0, B1} (Cai et al., 2023), ❹ SHViT {S1, S2, S3} (Yun & Ro, 2024), and ❺ MobileNetV4 {Conv-Small, Conv-Medium, Hybrid-Medium} (Qin et al., 2025). We compare these architectures with our high-speed ViT+Jumbo variants

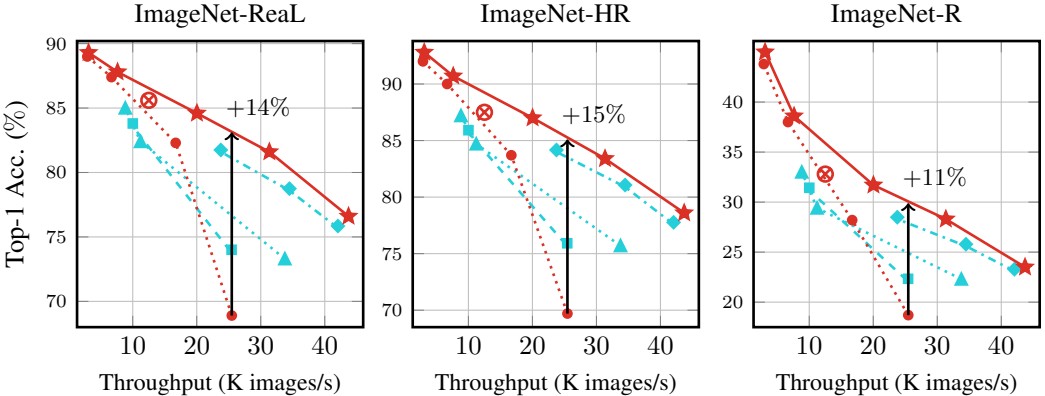

Legend: ★ViT+Jumbo (ours), ●ViT+Registers, ⊗BiXT, ▲MobileNetV4, ◆SHViT, ■EfficientViT

Figure 4: ViT+Jumbo achieves the Pareto frontier and is much simpler than specialized compute-efficient architectures. Results are plotted for each model's best learning rate. Throughput is measured on an RTX 4090 GPU using PyTorch 2.6.0, `torch.compile`, and a 512 batch size.

{Pico, Nano, Tiny, Small, Base}. Darcet et al. (2024) show ViT+Registers with $R$=16 performs best, which we confirm in the appendix Table 7 and use in these experiments. ViT+Jumbo is robust to the choice of $J$; we use $J$=3 for Base and $J$=6 otherwise.

**Test Sets.** We test all models on the three most common ImageNet-1K test sets: ImageNet-Val (Russakovsky et al., 2015), ImageNet-ReaL (Beyer et al., 2020), and ImageNet-v2 (Recht et al., 2019). To further evaluate generalization we also test all models on ImageNet-HR (Fuller et al., 2024), for its image diversity and high-quality annotations, and ImageNet-R (Hendrycks et al., 2021), for its out-of-distribution images.

**Results.** As illustrated in Fig 4, Jumbo achieves the Pareto frontier on ImageNet-1K. Crucially, Jumbo achieves these results while preserving the many advantages and simplicity of plain ViTs. Even matching the specialized compute-efficient architectures makes a strong case for ViT+Jumbo.

ViT+Jumbo outperforms ViT+Registers by 13% at the nano scale and 4% at the tiny scale, where such gains are significant. This confirms our first hypothesis that Jumbo's gains should increase as we decrease the patch width, i.e., from Small (384) to Tiny (192) to Nano (128) (Figs. 1a, 4).

ViT+Jumbo is a clear choice if a researcher or practitioner requires high speed and out-of-the-box ViT compatibility for SSL algorithms or multimodal processing. ViT+Registers is not as accurate at high speed, while the specialized compute-efficient architectures do not support most SOTA SSL algorithms or flexible processing across modalities. Remote sensing (Rolf et al., 2024) and autonomous driving (Muhammad et al., 2020) are two of many applications where this combination of speed, SSL support, and multimodal processing is particularly valuable.

## 4.2 IMAGENET-21K EXPERIMENTS WITH VIT COMPARISONS

ImageNet-1K is a subset of the more challenging, original ImageNet (Deng et al., 2009), now referred to as ImageNet-21K. We use a common variant comprising 10,450 classes that includes processing to make a more accessible benchmark (Ridnik et al., 2021). This dataset provides more than $10\times$ the number of classes and samples as ImageNet-1K, making it well suited to test our second hypothesis—that is, gains due to Jumbo should increase with increasing task-output dimensionality.

**Setup.** We train models from scratch on ImageNet-21K. Since training models on ImageNet-21K is expensive, we leverage a token dropping strategy to reduce costs. Specifically, we start training with a 90% token drop rate and linearly decrease this value to 10%; this halves the total number of tokens processed. Dehghani et al. (2024) demonstrate the effectiveness of this strategy, i.e., leveraging "masking" with plain supervised training. Plain ViTs support masking with minimal code changes.

We train each model architecture once, for 50 epochs using a 3e-3 learning rate and a 1024 batch size with the AdamW optimizer (Loshchilov, 2017) (see the Appendix A.4.1 for other hyperparameters).

**Baselines.** We choose ViT+Registers {Small, Base} to compare with our ViT+Jumbo {Small, Base} sizes. This is a more narrow but valid comparison, as the plain ViT is the vision community's preferred architecture at these scales. For ViT+Registers, we use $R$=16; for ViT+Jumbo, we use $J$=6 for the Small model and $J$=3 for the Base model.

**Results.** ViT+Jumbo outperforms ViT+Registers by 3.1% and 1.2% at ViT-Small and ViT-Base scales, respectively (see Fig. 1b). Gains due to Jumbo increase when scaling from ImageNet-1K to ImageNet-21K for a given model size, e.g., ViT-Small gains increase from 0.8% (Fig. 1a) to 3.1%. Thus, these findings confirm our second hypothesis that gains due to Jumbo should increase with increasing output dimensionality. Furthermore, for a given accuracy Jumbo is 1.9× faster (Fig. 1b).

### 4.3 SEMANTIC SEGMENTATION EXPERIMENTS

**Setup.** We fine-tune on ADE20K (Zhou et al., 2017) for 100 epochs with a batch size of 16 and sweep learning rates {1e-5, 2e-5, 4e-5, 8e-5} for each model size {ViT-Base, ViT-Small, ViT-Tiny} and global strategies {Jumbo, Registers}. We use 512×512 px images, which is standard for this dataset. We initialize model parameters from our ImageNet-1K supervised training experiments. We attach a Segmenter head (Strudel et al., 2021), which is a commonly used head for ViTs.

**Results.** Our Jumbo achieves mIoU of 44.4% / 39.1% / 35.5%, while Registers achieves mIoU of 42.5% / 37.0% / 32.4% for ViT-Base / Small / Tiny sizes. This shows Jumbo supports segmentation— since it is non-hierarchical—and *gains* when using a standard plain-ViT segmentation head.

### 4.4 MASKED AUTOENCODING EXPERIMENTS

**Setup.** We pre-train Jumbo ViT-Base and ViT-Large models using masked autoencoding (MAE; (He et al., 2022)) using the default settings. This tests Jumbo's ability in a standard SSL framework. This also tests Jumbo's scalability to larger models (up to ViT-Large) and longer training schedules (up to 1600 epochs on ImageNet-1K). These experiments are expensive, so we leverage free TPU resources, perform no hyperparameter tuning, and compare against plain ViT results obtained with the same MAE implementation. After pre-training, we linear probe on ImageNet-1K to obtain accuracies.

Table 1: **MAE Pretraining.** Jumbo significantly outperforms standard ViT. Jumbo scales to these large MAE models and their long training schedules (1600 epochs for ViT-Base, 800 epochs for ViT-Large). After pretraining, we linearly probe to compute top-1 accuracy on ImageNet-1K.

| Architecture | Speed K imgs/s | Params M | Memory GB | FLOPs G | Top-1 Acc. % |
|---|---|---|---|---|---|
| ViT-Base | 3.1 | 86.6 | 3.3 | 16.5 | 68.1 |
| ViT-Base+Jumbo | 3.1 | 130.7 | 3.9 | 16.9 | **73.0** |
| ViT-Large | 1.0 | 304.4 | 5.0 | 59.7 | 73.0 |
| ViT-Large+Jumbo | 1.0 | 382.2 | 5.2 | 59.9 | **74.0** |

**Results.** Our ViT-Base+Jumbo MAE outperforms the baseline by 4.9% on ImageNet-1K. ViT-**Base**+Jumbo *ties* the ViT-**Large** MAE, while Jumbo is 3× faster with only 0.43× the parameters. This shows Jumbo can be applied to SSL by MAE to improve performance without further modification. The role of masking in the MAE suggests that the wider Jumbo token stores more global information. For this MAE, Jumbo is a more efficient way to scale model parameters than the wider ViT.

### 4.5 ROBUSTNESS AND TEST-TIME ADAPTATION EXPERIMENTS

**Setup.** We measure robustness to corruption with and without adaptation. We follow SAR (Niu et al., 2023) exactly, swapping in ViT-S models with Registers or our Jumbo from Sec. 4.1, and measure robustness to 15 corruptions at the highest severity from ImageNet-C (Hendrycks & Dietterich, 2019).

**Results.** Jumbo is both more accurate than Registers (+0.8% on IN-Val) and more robust than Registers on corrupted data (+3.6% on IN-C). Test-time adaptation by SAR further increases the robustness gain to +5.2%. In principle test-time adaptation can apply to any architecture, but in

Table 2: **Test-Time Adaptation (TTA).** Jumbo improves plain-ViT robustness *without* TTA (avg. ↑3.6%) and *with* TTA (avg. ↑5.2%) on ImageNet-C. We follow SAR (Niu et al., 2023) and test across 15 shifts at the highest severity. Jumbo is directly compatible with SOTA methods designed for ViTs, for instant use without tuning, unlike highly-specialized architectures (MobileNet, SHViT, ...).

| Method | Gauss. | Shot | Impul. | Defoc. | Glass | Motion | Zoom | Snow | Frost | Fog | Brit. | Contr. | Elastic | Pixel | JPEG | Avg. |
|---|---|---|---|---|---|---|---|---|---|---|---|---|---|---|---|---|
| Registers | 13.6 | 14.2 | 13.0 | 29.3 | 20.3 | 34.9 | 28.7 | 49.0 | 50.4 | 56.6 | 73.5 | 47.8 | 29.0 | 45.6 | 56.1 | 37.5 |
| Jumbo | **25.9** | **26.6** | **25.8** | **31.2** | **21.4** | **33.3** | **30.6** | **53.1** | **51.9** | **57.3** | **75.2** | **49.4** | **30.5** | **47.5** | **57.1** | **41.1** |
| Registers+SAR | 38.9 | 39.2 | 41.5 | 48.2 | 48.7 | 56.5 | 32.1 | 62.0 | 59.4 | 68.9 | 76.1 | 61.5 | 59.1 | 65.5 | 66.1 | 54.9 |
| Jumbo+SAR | **45.2** | **49.6** | **51.2** | **53.3** | **53.2** | **61.1** | **44.5** | **66.2** | **58.5** | **71.7** | **77.5** | **67.1** | **65.2** | **69.0** | **69.0** | **60.1** |

practice methods specialize. SOTA methods such as SAR are designed for the plain ViT LayerNorm, and not the BatchNorm of SHViT, MobileNetV4, and EfficientViT, so ViT compatibility is a plus.

## 4.6 TIME SERIES EXPERIMENTS

Jumbo can easily process different input shapes (beyond images) because it maintains the plain transformer interface. We apply Jumbo to time series inputs. PatchTST (Nie et al., 2023) is a SOTA patch-based transformer for time series that we extend with registers (PatchTST+Registers) or Jumbo (PatchTST+Jumbo).

**Setup.** We train models from scratch on ❶ 10 univariate time series datasets from the UCR archive (Dau et al., 2018), and ❷ 10 multivariate time series datasets from the UEA archive (Bagnall et al., 2018); both of which are commonly used benchmarks (Zerveas et al., 2021; Grover et al., 2024; Le et al., 2024). For each dataset and model, we perform a hyperparameter sweep from the Cartesian product of learning rate {3e-3, 1e-3, 3e-4, 1e-4}, and dropout {0.0, 0.1, 0.2}. More details are in Appendix A.4.2. We report the best run and the average of all 12 runs per experiment in the appendix Tables 12 & 13. To summarize these results, we compute the rank between models and then average the ranks over the 10 univariate and 10 multivariate datasets.

**Baselines.** We compare PatchTST with our PatchTST+Jumbo method and our PatchTST+Registers baseline. We experiment with 8 and 42 patches per sequence for all three models. Jumbo and registers are both simple to adopt for PatchTST because they remain plain transformers.

Table 3: **Time series** rankings using PatchTST (Nie et al., 2023) with Registers or Jumbo (*lower is better* and the best is in bold). We rank over 10 univariate and 10 multivariate datasets. "Best" is the best run of our 12-run hyperparameter sweep and "Avg" is the average over the sweep. Jumbo achieves the best ranking in all experiments. We use two patch sizes: 8/42 (results are formatted likewise).

| | | PatchTST | PatchTST +Registers | PatchTST +Jumbo |
|---|---|---|---|---|
| Univar. | Best | 2.0/1.9 | 2.5/2.1 | **1.5/1.7** |
| | Avg | 2.9/2.3 | 2.1/2.4 | **1.0/1.3** |
| Multivar. | Best | 2.1/2.0 | 2.1/1.9 | **1.6/1.7** |
| | Avg | 2.7/2.6 | 2.0/2.4 | **1.3/1.0** |

**Results.** PatchTST+Jumbo outperforms strong PatchTST and PatchTST+Registers baselines (Tab. 3). Jumbo gains the most with fewer patches and when considering overall results across hyperparameters. These results establish that Jumbo can improve non-causal transformers beyond ViTs.

## 4.7 PRELIMINARY LANGUAGE EXPERIMENTS

We perform proof-of-concept experiments on a vision-language task (image-caption retrieval) and a language-only task (masked modeling) to show that Jumbo may help natural language processing (NLP) tasks. These experiments are not meant to achieve SOTA results—we leave comprehensive NLP experiments for future work.

**Image-caption retrieval.** We align our Jumbo and Register models (ViT-Base size) to language embeddings by fine-tuning to predict image-caption embeddings given images. We pick the Qwen3 embedding model (Zhang et al., 2025) and fine-tune ViTs for 500K steps with a 128 batch size on the CC3M dataset (Changpinyo et al., 2021). For evaluation, we embed all captions in CC3M's validation set with Qwen3 and embed all corresponding images with our fine-tuned ViTs. Jumbo

image embeddings retrieve the correct caption embedding $34\%$ of the time in the top 10, whereas Register image embeddings score $30\%$.

**Masked language modeling (MLM).** We train Jumbo and Register models (Base size) from scratch using BERT's MLM objective and hyperparameters (Devlin et al., 2019). We pick the C4 dataset (Raffel et al., 2020) and train for 500K steps with a 128 batch size. Jumbo achieves a validation perplexity of 4.8, while Registers achieves 4.9 (lower is better). MLM perplexity correlates with performance on downstream tasks.

## 4.8 ABLATIONS

Table 4: Jumbo's shared FFN increases accuracy and is memory efficient. Our Jumbo FFN can be enlarged ($J$=10) for even higher performance, at relatively low cost. We report top-1 accuracy on ImageNet-21K.

| Architecture | Speed K imgs/s ↑ | Params M | Memory GB ↓ | FLOPs G ↓ | Top-1 Acc. % ↑ |
|---|---|---|---|---|---|
| Jumbo (Fig. 1b) | 7.9 | 88 | 2.6 | 4.6 | 44.61 |
| Jumbo without layer sharing | 7.7 | 557 | 4.1 | 4.6 | 44.95 |
| Jumbo without Jumbo FFN | 8.4 | 46 | 2.2 | 4.4 | 43.64 |
| Jumbo with LoRA, rank=8 | 7.7 | 89 | 2.5 | 4.6 | 44.94 |
| Jumbo $J$: 6→10 | 6.9 | 180 | 3.4 | 5.5 | 45.62 |

**Setup.** We follow our ImageNet-21K training recipe and ablate Jumbo's design at ViT-Small scale to better understand the contributions of the architecture and its design choices.

**Results.** (Tab. 4) Not sharing the Jumbo FFN across layers slightly improves accuracy at this scale. However, we can fully recover from the drop with sharing by adapting the Jumbo FFN parameters with LoRAs (Hu et al., 2022): we still share the Jumbo FFN across layers but apply layer-specific LoRAs to specialize efficiently. LoRAs recover accuracy at negligible cost in speed and memory. Jumbo without Jumbo FFNs performs well enough ($2.2\%$ better than ViT+Registers) but worse than Jumbo: the main difference between this ablation and ViT+Registers is that it concatenates all global tokens as input to the classifier (rather than discarding registers). Yet, our best ViT-Small includes the Jumbo FFN: with $J$=10 its shared FFN achieves $45.6\%$ top-1 accuracy. This Jumbo model beats ViT-Small+Registers by $4.1\%$ and matches ViT-**Base**+Registers ($0.1\%$ difference) with higher speed ($2.4\times$ faster) and less memory.

## 4.9 ANALYSIS: HOW TO SCALE EFFICIENCY AND CAPACITY

Table 5: ViT-Base+Jumbo matches a *symmetrically wider* ViT+Registers with equal params; yet our Jumbo is $1.7\times$ faster. Jumbo also outperforms other ways of adding global capacity, e.g., ❶ uses an FFN for patches, and a separate FFN for CLS+Reg. tokens, ❷ uses an FFN for patches+CLS, and separate FFN for Reg. tokens. We report top-1 accuracy on ImageNet-21K.

| Architecture | Speed K imgs/s ↑ | Params M | Memory GB ↓ | FLOPs G ↓ | Top-1 Acc. % ↑ |
|---|---|---|---|---|---|
| **ViT-Base models** | | | | | |
| Jumbo | 3.1 | 152 | 4.1 | 16.5 | 47.0 |
| Reg. (Darcet et al., 2024) | 2.9 | 94 | 3.5 | 18.2 | 45.7 |
| Registers $D$: 768→1024 | 1.8 | 163 | 4.5 | 32.4 | 47.1 |
| Alt. ❶: CLS+Reg. FFN | 2.8 | 151 | 4.1 | 18.2 | 46.0 |
| Alt. ❷: Reg. FFN | 2.8 | 151 | 4.1 | 18.2 | 46.4 |
| **ViT-Small models** | | | | | |
| Jumbo | 7.9 | 88 | 2.6 | 4.6 | 44.6 |
| Reg. (Darcet et al., 2024) | 8.0 | 26 | 2.3 | 4.6 | 41.5 |
| Alt. ❶: CLS+Reg. FFN | 7.7 | 40 | 2.3 | 4.6 | 41.5 |
| Alt. ❷: Reg. FFN | 7.7 | 40 | 2.3 | 4.6 | 42.1 |

**Is Jumbo more accurate just because it has more parameters? No.** We take ViT-B+Registers and increase its width 768→1024 to equalize the number of parameters with our ViT-Base+Jumbo (Tab. 5 rows 1 & 3). These models differ in accuracy by $0.1\%$, yet Jumbo is more efficient with $1.7\times$ the throughput, $0.5\times$ the FLOPs, and $0.9\times$ the memory. Our novel asymmetric-width design of the Jumbo token and FFN is crucial to its better efficiency.

**Alternate ViT+Register designs.** We experiment with two more architectures to investigate the role of adding separate FFNs for different types of tokens (Tab. 5). Alternative ❶ has an FFN for all patch tokens with a *separate* FFN for the CLS and registers. Alternative ❷ has an FFN for all patch tokens

*and* the `CLS` token with a separate FFN for the registers. Neither model gains much: the asymmetric token width of Jumbo explains its success, and *not* the addition of more parameters alone.

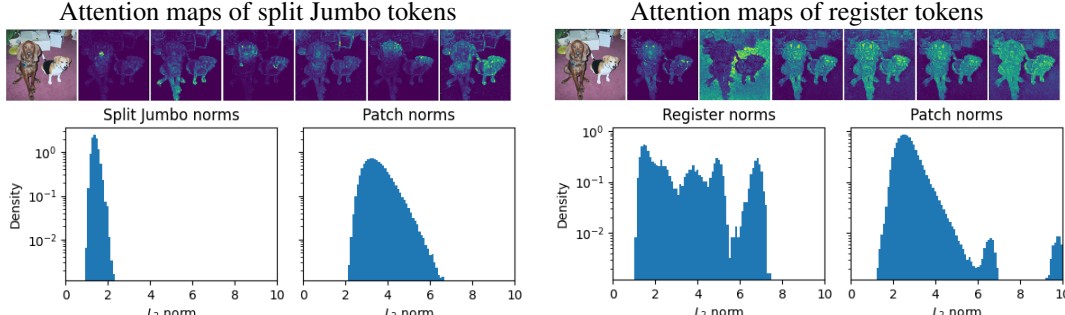

Figure 5: Jumbo (left two subfigures) eliminates high-norm, outlier tokens in our measurements. According to Darcet et al. (2024), outlier tokens cause attention-map artifacts, and their presence can be reduced by adding registers (right two subfigures). By inspection, Jumbo also learns artifact-free attention maps, and split Jumbo tokens seem to specialize.

**Does Jumbo also reduce high-norm tokens?** Registers reduce high-norm, outlier tokens that cause attention map artifacts (Darcet et al., 2024). We test if Jumbo does the same. The ViT+Jumbo models we train are in fact *more effective* at reducing outlier tokens than ViT+Registers (Fig. 5). We also show attention maps in the Appendix A.7 where we again see a similar effect.

## 5 DISCUSSION: EFFICIENCY, GENERALITY, AND CAPACITY

**Limitations and Future Work.** In this work, we conduct only proof-of-concept experiments using Jumbo in vision-language (e.g., CLIP (Radford et al., 2021b)) and language-only applications (e.g., BERT (Devlin et al., 2019), which is non-causal and could benefit from Jumbo in theory). We save further exploring these applications for future work.

**Conclusion.** Jumbo is highly efficient, simple, and general: our Jumbo ViTs achieve SOTA accuracy-speed trade-offs by a targeted increase in the global computation and parameter capacity of any plain ViT. We show that upgrading a plain ViT with Jumbo improves accuracy at the same speed or maintains accuracy at faster speeds for supervised image classification, image segmentation, self-supervised learning, time series modeling, and test-time adaptation. Jumbo is the first attention-only and non-hierarchical architecture to outperform specialized compute-efficient architectures like EfficientViT (Cai et al., 2023). To do so Jumbo increases width *asymmetrically*, across tokens, and not across layers (in contrast to existing hierarchical models). While increasing model capacity can increase accuracy, it is critical to add capacity in the right places to *achieve high efficiency and maintain model flexibility* as we show with Jumbo.

## ACKNOWLEDGEMENTS

AF is primarily supported by an NSERC PGS-D scholarship. ES is supported by a Canada CIFAR AI Chair. YY is primarily supported by an Ontario Graduate Scholarship (OGS) and a Vector Scholarship in AI. Resources used in preparing this research were provided, in part, by the Province of Ontario, the Government of Canada through CIFAR, and companies sponsoring the Vector Institute. We thank the Google TPU Research Cloud (TRC) for providing the TPUs on which we conducted the MAE experiments. We thank Eric Tzeng and Max Argus for their helpful review and feedback on the exposition.

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

# A    APPENDIX

## A.1    IMPACT STATEMENT

This work presents new designs and empirical results for deep network architectures for more accurate and computationally efficient modeling applied to visual recognition and time series processing. This general topic does not have more specific societal consequences aside from those inherited, good or bad, from the adoption of machine learning.

## A.2    COMPUTE-EFFICIENT ARCHITECTURE DESCRIPTIONS

❶ EfficientViT (Cai et al., 2023) is a hierarchical architecture with four stages and one head. Stages 1 and 2 consist of MBConv layers (Sandler et al., 2018). Stages 3 and 4 consist of MBConv sublayers and their novel EfficientViT sublayer, consisting of an efficient attention module and an FFN+DWConv module (Howard, 2017). Their attention module creates queries, keys, and values of three scales via three DWConvs, and then each set of queries, keys, and values undergoes efficient linear attention. Finally, the head receives outputs from Stages 2, 3, and 4, and applies a final MBConv. EfficientViT variants differ in stage depths and widths, as well as head width.

❷ SHViT (Yun & Ro, 2024) is a hierarchical architecture with three stages. Stage 1 consists of a DWConv+BatchNorm sublayer and an FFN sublayer. Stages 2 and 3 incorporate their novel single-headed self-attention (SHSA) sublayer between the stage 1 sublayers. SHSA consists of performing single-headed self-attention on a fraction of dimensions ($1/4.67$ ratio); the other dimensions pass straight through, further reducing cost. Both FFN and SHSA sublayers also replace linear layers with DWConv. SHViT variants differ in stage depths and widths.

❸ MobileNetV4 (Qin et al., 2025) variants use their FusedIB, ExtraDW, and Mobile MQA (multi-query attention) modules along with MBConv, ConvNext-Like (Liu et al., 2022), and FFN modules. Variants differ in stage depths and widths, the number of stages, and stage architectures built with a combination of the listed modules.

## A.3    DISCUSSION OF SEQUENCE-BASED ARCHITECTURES

Both Vim (Zhu et al., 2024) and V-RWKV (Duan et al., 2024) are not ViTs nor attention-only; thus, they require further customization to process other shapes. Plain ViTs simply require defining position embeddings to accommodate new shapes, which is trivial (e.g., one line of code).

Plain ViTs (including with Registers or Jumbo) are flexible to input shapes:

(1) ViTs can drop/mask tokens to process any subset of patches simply via indexing, which is not possible for other architectures. For example, SHViT, MobileNetV4, and EfficientViT all use convolutions and pooling. Mamba-based architectures, such as Vim, require defining a patch-order, e.g., left-to-right. Dropping patches in such a sequence model incorrectly assigns their positions, unlike transformers. Out of the box, V-RWKV's "Q-Shift" does not support token dropping since it is convolutional, but the authors of V-RWKV modify it to enable MAE pre-training. Token dropping/masking is used in all SOTA SSL algorithms, can speed up training and inference (e.g., through adaptive computation), and is thus a desirable trait of model architectures.

(2) ViTs can process non-2D data, for example, remote sensing inputs or combinations of different shapes. SOTA remote sensing foundation models process inputs of shape: height $\times$ width $\times$ spectral groups $\times$ time (Tseng et al., 2025). Applying Mamba-based architectures to this input would require defining complex (and unnatural) scans over each axis. Applying V-RWKV to this input would require re-designing the Q-Shift operation. Plain ViTs require defining position embeddings, which is trivial and explains their dominance in remote sensing and other applications. Plain ViTs can also process inputs composed of different shapes, e.g., a joint language-vision encoder, via position embeddings. SHViT, MobileNetV4, EfficientViT, Vim, and V-RWKV would all require significant customization to jointly encode 1D (language) and 2D (vision) inputs.

## A.4 Experimental Details

### A.4.1 ImageNet-1K and -21K Hyperparameters

We pick these recipes based on findings in the literature—such as (Touvron et al., 2022), (Fuller et al., 2024), (Beyer et al., 2022), (Dehghani et al., 2024), and (Steiner et al., 2022)—and past experience indicating that these recipes would result in strong models.

**ImageNet-1K training recipe:** $128 \times 128$ px images, 400 epochs, 1024 batch size, PyTorch's AdamW optimizer with a 0.05 weight decay, 1.0 clip grad norm, `deit3-base-patch16-224.fb-in22k-ft-in1k` teacher (Touvron et al., 2022) given $224 \times 224$ px images using (Wightman, 2019)'s implementation, KL divergence loss between student and teacher logits (Beyer et al., 2022), linear learning rate warmup for 10% of steps to $\{1\text{e-}3, 3\text{e-}3\}$ and cooldown using a cosine decay schedule to 1e-5, mixup $\alpha = 0.8$, cutmix $\alpha = 1$, and 3-Augment data augmentation (Touvron et al., 2022). Then we continue training at $224 \times 224$ px images, 20 epochs, 512 batch size, PyTorch's AdamW optimizer with a 0.1 weight decay, 1.0 clip grad norm, `deit3-large-patch16-224.fb-in22k-ft-in1k` teacher (Touvron et al., 2022) given $224 \times 224$ px images using (Wightman, 2019)'s implementation, KL divergence loss between student and teacher logits (Beyer et al., 2022), linear learning rate warmup for 25% of steps to 5e-5 and cooldown using a cosine decay schedule to 1e-5, mixup $\alpha = 0.8$, cutmix $\alpha = 1$, and AutoAugment ("rand-m9-mstd0.5-inc1") data augmentation (Cubuk et al., 2018) following DEIT III's (Touvron et al., 2022) high-res finetuning recipe.

**ImageNet-21K training recipe:** $224 \times 224$ px images, 50 epochs, 1024 batch size, PyTorch's AdamW optimizer with a 0.02 weight decay, 1.0 clip grad norm, cross-entropy loss, linear learning rate warmup for 10% of steps to 3e-3 and cooldown using a cosine decay schedule to 1e-5, mixup $\alpha = 0.8$, cutmix $\alpha = 0$, and 3-Augment data augmentation (Touvron et al., 2022). To speed up training, we also employ a token dropping strategy starting at 90%, linearly decreasing to 10%.

### A.4.2 Time Series Experiments

We adopt the PatchTST (Nie et al., 2023) architecture for our time series experiments. PatchTST is a patch-based transformer architecture for time series processing. The method splits a univariate time series into patches processed as they are in ViTs for classification, aside from position encoding (2D vs. 1D). For multivariate series, each channel is processed *independently* using the shared transformer backbone, with the final-layer `CLS` tokens from each channel concatenated before classification. We extend this shared backbone with registers (PatchTST+Registers) and Jumbo (PatchTST+Jumbo).

We closely follow the PatchTST training recipe for our experiments, making minor adjustments based on prior experience to enhance performance. This method remains competitive with recent transformer-based benchmarks for time series classification (Zerveas et al., 2021; Grover et al., 2024; Le et al., 2024). Apart from variations in time series length, all experiments use the same hyperparameters and methodology.

**PatchTST Hyperparameters:** The model comprises 3 encoder layers, each with 16 attention heads and a token width of $D = 128$. The transformer FFN includes two linear layers with a GELU activation (Hendrycks & Gimpel, 2016); the first expands the hidden dimension to 256, while the second projects it back to 128. For PatchTST+Jumbo, we use $J = 4$. For PatchTST+Registers, $R$ is calculated to match FLOPs.

**Time Series training recipe:** We perform a hyperparameter sweep over the Cartesian product of learning rates $\{3\text{e-}3, 1\text{e-}3, 3\text{e-}4, 1\text{e-}4\}$ and dropout rates $\{0.0, 0.1, 0.2\}$. Each configuration uses either 8 or 42 equally sized patches of maximum possible length, with end-padding applied as needed. The stride length is set to half the patch length. Unless stated otherwise, all experiments follow the same setup: 100 epochs, 256 batch size, PyTorch's AdamW optimizer with a 0.02 weight decay, cross-entropy loss, and a linear learning rate warmup for the first 10% of steps, followed by a cooldown using cosine decay to 1e-8. For large datasets, we reduce the number of epochs to ensure efficient processing within a reasonable time frame; specifically, we train datasets {Sleep, Tiselac, FaceDetection} for 20 epochs.

Each dataset from the UEA and UCR archives includes a prescribed validation set. We create a new 50/50 test/validation split from each of these original validation sets, selecting the best run based on validation performance. All reported results are from the *test* set.

The 20 datasets were selected in decreasing order of their number of training examples; datasets with either (i) fewer than 42 total timesteps or (ii) significant data preparation issues were excluded.

## A.5 DETAILED IMAGENET-1K RESULTS

Table 6: All final results obtained on $224 \times 224$ px images (%).

| Architecture | Size | Throughput $224^2$ px | ImageNet-Val Top-1 | Top-5 | ImageNet-ReaL Top-1 | Top-5 | ImageNet-v2 Top-1 | Top-5 | ImageNet-R Top-1 | Top-5 | ImageNet-HR Top-1 | Top-5 |
|---|---|---|---|---|---|---|---|---|---|---|---|---|
| ViT+Jumbo | $D$=96, $J$=6 | 43.7K | 69.1 | 88.5 | 76.6 | 92.3 | 56.2 | 79.2 | 21.9 | 35.0 | 78.6 | 92.8 |
| | $D$=128, $J$=6 | 31.3K | 74.6 | 91.9 | 81.6 | 94.9 | 61.6 | 83.6 | 28.3 | 43.0 | 83.4 | 94.9 |
| | $D$=192, $J$=6 | 20.4K | 78.4 | 94.0 | 84.6 | 96.2 | 66.1 | 86.6 | 31.7 | 46.7 | 87.0 | 96.7 |
| | $D$=384, $J$=6 | 7.6K | 82.6 | 96.5 | 87.8 | 97.9 | 71.8 | 90.8 | 38.6 | 54.9 | 90.7 | 98.2 |
| | $D$=768, $J$=3 | 3.1K | 84.9 | 97.5 | 89.3 | 98.5 | 75.7 | 93.0 | 45.0 | 62.4 | 92.8 | 99.0 |
| ViT+Registers | $D$=128, $R$=16 | 25.5K | 61.0 | 84.1 | 68.9 | 88.9 | 49.1 | 74.1 | 18.7 | 32.4 | 69.7 | 88.9 |
| | $D$=192, $R$=16 | 16.7K | 74.5 | 92.3 | 82.3 | 95.4 | 62.5 | 84.7 | 28.2 | 43.5 | 83.7 | 95.6 |
| | $D$=384, $R$=16 | 6.6K | 81.9 | 96.0 | 87.4 | 97.7 | 71.4 | 90.2 | 38.0 | 53.8 | 90.0 | 98.0 |
| | $D$=768, $R$=16 | 2.9K | 84.3 | 97.2 | 89.0 | 98.3 | 74.8 | 92.5 | 43.8 | 60.7 | 92.0 | 98.9 |
| MobileNetV4 | conv-small | 33.7K | 65.6 | 86.2 | 73.3 | 90.7 | 52.5 | 75.5 | 22.3 | 37.5 | 75.7 | 91.3 |
| | conv-medium | 11.2K | 74.9 | 92.6 | 82.4 | 95.5 | 63.1 | 84.8 | 29.4 | 45.9 | 84.7 | 95.8 |
| | hybrid-medium | 8.8K | 78.1 | 94.3 | 85.0 | 96.7 | 67.0 | 87.5 | 33.0 | 49.4 | 87.2 | 96.8 |
| SHViT | S1 | 42.1K | 67.9 | 88.2 | 75.7 | 92.2 | 54.7 | 78.2 | 23.1 | 38.0 | 77.6 | 92.7 |
| | S2 | 34.5K | 71.0 | 90.0 | 78.6 | 93.6 | 58.4 | 80.5 | 25.6 | 41.1 | 80.9 | 93.9 |
| | S3 | 23.7K | 74.3 | 92.0 | 81.6 | 95.0 | 61.8 | 83.5 | 28.3 | 43.9 | 84.0 | 95.4 |
| EfficientViT | B0 | 25.2K | 66.3 | 86.5 | 73.9 | 90.7 | 53.6 | 76.3 | 22.2 | 36.7 | 75.8 | 91.0 |
| | B1 | 9.8K | 76.9 | 93.5 | 83.7 | 96.2 | 64.5 | 85.9 | 31.3 | 47.2 | 85.8 | 96.4 |

Table 7: ViT+Registers results, obtained on $128 \times 128$ px images (%).

| Patch Width | Num. Registers | Learning Rate | Throughput imgs/s | ImageNet-Val Top-1 | Top-5 | ImageNet-ReaL Top-1 | Top-5 | ImageNet-v2 Top-1 | Top-5 | ImageNet-R Top-1 | Top-5 | ImageNet-HR Top-1 | Top-5 |
|---|---|---|---|---|---|---|---|---|---|---|---|---|---|
| 128 | 16 | 3e−3 | 107.0K | 53.6 | 78.5 | 60.8 | 83.6 | 42.4 | 67.8 | 15.9 | 28.6 | 61.9 | 83.4 |
| | 16 | 1e−3 | | 51.9 | 76.8 | 59.0 | 81.8 | 40.8 | 65.4 | 13.7 | 24.9 | 60.5 | 82.6 |
| 192 | 8 | 3e−3 | 65.7K | 68.5 | 88.8 | 76.1 | 92.6 | 55.7 | 78.8 | 24.9 | 39.4 | 77.8 | 92.2 |
| | 16 | 3e−3 | 59.9K | 68.8 | 88.9 | 76.6 | 92.6 | 55.9 | 79.4 | 24.8 | 38.9 | 78.5 | 92.5 |
| | 16 | 1e−3 | | 66.1 | 87.2 | 74.0 | 91.2 | 54.2 | 77.0 | 22.9 | 36.2 | 75.4 | 91.6 |
| 384 | 8 | 3e−3 | 24.6K | 77.8 | 93.9 | 84.3 | 96.2 | 65.8 | 86.3 | 33.3 | 48.6 | 86.8 | 96.5 |
| | 16 | 3e−3 | 21.8K | 78.1 | 94.0 | 84.5 | 96.3 | 66.1 | 86.6 | 33.3 | 48.6 | 86.6 | 96.5 |
| | 16 | 1e−3 | | 78.2 | 94.1 | 84.5 | 96.3 | 66.4 | 86.6 | 33.5 | 48.4 | 87.2 | 96.6 |

Table 8: ViT+Jumbo results, obtained on $128 \times 128$ px images (%).

| Patch Width | Learning Rate | Throughput imgs/s | ImageNet-Val Top-1 | Top-5 | ImageNet-ReaL Top-1 | Top-5 | ImageNet-v2 Top-1 | Top-5 | ImageNet-R Top-1 | Top-5 | ImageNet-HR Top-1 | Top-5 |
|---|---|---|---|---|---|---|---|---|---|---|---|---|
| 96 | 3e−3 | 136.0K | 62.4 | 83.7 | 69.6 | 88.1 | 49.3 | 72.9 | 20.1 | 33.5 | 71.4 | 89.1 |
| | 1e−3 | | 60.8 | 82.9 | 68.0 | 87.3 | 48.3 | 71.8 | 18.9 | 31.6 | 70.3 | 87.6 |
| 128 | 3e−3 | 103.1K | 67.7 | 87.5 | 75.0 | 91.2 | 54.3 | 77.6 | 24.1 | 37.5 | 76.6 | 92.1 |
| | 1e−3 | | 68.4 | 87.9 | 75.6 | 91.6 | 55.2 | 78.0 | 23.8 | 37.6 | 77.0 | 92.1 |
| 192 | 3e−3 | 57.3K | 73.3 | 91.2 | 80.2 | 94.1 | 60.5 | 82.1 | 28.0 | 42.2 | 82.7 | 94.6 |
| | 1e−3 | | 73.5 | 91.3 | 80.3 | 94.1 | 60.5 | 81.8 | 27.8 | 42.2 | 82.8 | 94.4 |
| 384 | 3e−3 | 20.4K | 79.3 | 94.4 | 85.3 | 96.5 | 67.3 | 87.0 | 34.3 | 49.5 | 88.3 | 96.8 |
| | 1e−3 | | 79.3 | 94.5 | 85.1 | 96.6 | 66.7 | 86.7 | 33.4 | 48.4 | 87.7 | 96.6 |

Table 9: MobileNetV4 results, obtained on $128 \times 128$ px images (%).

| Size | Learning Rate | Throughput imgs/s | ImageNet-Val Top-1 | Top-5 | ImageNet-ReaL Top-1 | Top-5 | ImageNet-v2 Top-1 | Top-5 | ImageNet-R Top-1 | Top-5 | ImageNet-HR Top-1 | Top-5 |
|---|---|---|---|---|---|---|---|---|---|---|---|---|
| conv-small | 3e−3 | 142.7K | 62.1 | 83.6 | 69.2 | 88.1 | 49.1 | 72.8 | 20.4 | 34.3 | 71.8 | 89.4 |
|  | 1e−3 |  | 60.0 | 82.0 | 67.2 | 86.7 | 47.6 | 71.4 | 18.9 | 32.2 | 69.8 | 87.7 |
| conv-medium | 3e−3 | 53.8K | 73.3 | 91.5 | 80.5 | 94.6 | 60.6 | 82.9 | 27.7 | 42.8 | 83.2 | 95.3 |
|  | 1e−3 |  | 72.2 | 90.7 | 79.4 | 94.0 | 59.5 | 81.7 | 27.0 | 42.0 | 82.0 | 94.6 |
| hybrid-medium | 3e−3 | 43.5K | 74.9 | 92.4 | 81.8 | 95.3 | 62.4 | 84.0 | 29.5 | 44.8 | 84.4 | 95.5 |
|  | 1e−3 |  | 75.2 | 92.5 | 82.0 | 95.3 | 63.0 | 84.5 | 29.1 | 44.5 | 84.2 | 95.4 |

Table 10: SHViT results, obtained on $128 \times 128$ px images (%).

| Size | Learning Rate | Throughput imgs/s | ImageNet-Val Top-1 | Top-5 | ImageNet-ReaL Top-1 | Top-5 | ImageNet-v2 Top-1 | Top-5 | ImageNet-R Top-1 | Top-5 | ImageNet-HR Top-1 | Top-5 |
|---|---|---|---|---|---|---|---|---|---|---|---|---|
| S1 | 3e−3 | 81.0K | 63.5 | 84.9 | 70.9 | 89.1 | 50.8 | 74.5 | 22.2 | 35.7 | 73.7 | 90.0 |
|  | 1e−3 |  | 63.5 | 85.1 | 71.0 | 89.3 | 50.9 | 74.4 | 21.3 | 34.7 | 72.9 | 90.5 |
| S1 | 3e−3 | 76.1K | 66.6 | 87.0 | 73.9 | 90.8 | 54.0 | 76.6 | 23.9 | 38.0 | 76.1 | 91.7 |
|  | 1e−3 |  | 66.7 | 87.0 | 73.8 | 90.8 | 53.7 | 76.8 | 24.0 | 37.8 | 76.7 | 92.0 |
| S3 | 3e−3 | 73.8K | 70.5 | 89.8 | 77.7 | 93.1 | 58.1 | 80.4 | 26.6 | 41.0 | 80.4 | 93.8 |
|  | 1e−3 |  | 71.2 | 90.0 | 78.3 | 93.3 | 58.6 | 80.7 | 26.7 | 40.7 | 80.7 | 93.9 |

Table 11: EfficientViT results, obtained on $128 \times 128$ px images (%).

| Size | Learning Rate | Throughput imgs/s | ImageNet-Val Top-1 | Top-5 | ImageNet-ReaL Top-1 | Top-5 | ImageNet-v2 Top-1 | Top-5 | ImageNet-R Top-1 | Top-5 | ImageNet-HR Top-1 | Top-5 |
|---|---|---|---|---|---|---|---|---|---|---|---|---|
| B0 | 3e−3 | 98.6K | 59.5 | 81.9 | 66.8 | 86.7 | 46.8 | 70.3 | 18.6 | 32.0 | 69.3 | 87.6 |
|  | 1e−3 |  | 60.8 | 82.6 | 68.0 | 87.2 | 48.3 | 71.6 | 19.3 | 32.6 | 70.4 | 87.7 |
| B1 | 3e−3 | 38.7K | 71.8 | 90.7 | 79.2 | 94.0 | 59.7 | 81.8 | 27.4 | 42.2 | 81.5 | 94.4 |
|  | 1e−3 |  | 72.8 | 91.0 | 79.8 | 94.2 | 60.4 | 81.9 | 27.1 | 42.3 | 82.5 | 94.8 |

## A.6 DETAILED TIMESERIES RESULTS

Table 12: Univariate time series classification results (%). "Best" refers to the best run of our 12-run hyperparameter sweep and "Avg" refers to the average over the sweep.

|  |  | PatchTST/8 | PatchTST/8 +Registers | PatchTST/8 +Jumbo | PatchTST/42 | PatchTST/42 +Registers | PatchTST/42 +Jumbo |
|---|---|---|---|---|---|---|---|
| Sleep | Best | 70.9 | 70.7 | 73.3 | 70.5 | 70.6 | 70.3 |
|  | Avg | 67.5 | 67.7 | 68.3 | 67.2 | 67.1 | 67.6 |
| InsectSound | Best | 82.8 | 83.3 | 83.7 | 85.8 | 84.4 | 85.6 |
|  | Avg | 76.7 | 76.0 | 78.7 | 78.7 | 78.7 | 79.7 |
| FruitFlies | Best | 92.2 | 90.9 | 92.2 | 95.2 | 95.0 | 95.1 |
|  | Avg | 88.4 | 88.4 | 89.4 | 93.1 | 92.9 | 93.9 |
| RightWhaleCalls | Best | 94.3 | 93.8 | 95.1 | 96.7 | 97.0 | 96.1 |
|  | Avg | 92.8 | 93.5 | 94.2 | 94.0 | 94.8 | 95.1 |
| FaultDetectionA | Best | 98.0 | 97.7 | 98.1 | 99.6 | 99.8 | 99.8 |
|  | Avg | 94.6 | 95.2 | 97.2 | 99.2 | 99.2 | 99.5 |
| ElectricDevices | Best | 89.0 | 88.8 | 90.1 | 92.4 | 92.4 | 92.5 |
|  | Avg | 81.6 | 83.2 | 84.0 | 85.1 | 85.2 | 88.1 |
| Crop | Best | 80.9 | 81.2 | 82.0 | 81.2 | 80.6 | 82.2 |
|  | Avg | 69.4 | 70.9 | 72.0 | 68.7 | 68.3 | 68.7 |
| FordB | Best | 98.8 | 97.3 | 97.7 | 97.7 | 96.5 | 96.5 |
|  | Avg | 95.4 | 96.0 | 96.6 | 95.7 | 94.9 | 94.8 |
| FordA | Best | 97.3 | 98.0 | 97.7 | 97.1 | 97.1 | 97.5 |
|  | Avg | 96.1 | 96.6 | 97.2 | 95.5 | 95.7 | 96.0 |
| MelbournePedestrian | Best | 92.1 | 91.5 | 91.0 | 90.4 | 91.0 | 93.1 |
|  | Avg | 81.8 | 82.8 | 83.9 | 83.5 | 83.8 | 84.9 |

Table 13: Multivariate time series classification results (%). "Best" refers to the best run of our 12-run hyperparameter sweep and "Avg" refers to the average over the sweep.

|  |  | PatchTST/8 | PatchTST/8 +Registers | PatchTST/8 +Jumbo | PatchTST/42 | PatchTST/42 +Registers | PatchTST/42 +Jumbo |
|---|---|---|---|---|---|---|---|
| Tiselac | Best | 96.6 | 96.9 | 97.2 | 96.4 | 96.4 | 96.7 |
|  | Avg | 86.9 | 87.8 | 90.1 | 84.7 | 85.0 | 87.9 |
| WalkingSittingStanding | Best | 96.0 | 96.0 | 96.5 | 98.0 | 97.6 | 97.6 |
|  | Avg | 91.7 | 89.8 | 93.5 | 93.9 | 93.9 | 94.5 |
| SpokenArabicDigits | Best | 99.9 | 99.7 | 99.9 | 99.7 | 99.9 | 99.9 |
|  | Avg | 99.5 | 99.6 | 99.6 | 99.6 | 99.5 | 99.7 |
| FaceDetection | Best | 87.8 | 88.1 | 87.5 | 86.8 | 86.6 | 84.8 |
|  | Avg | 78.9 | 80.9 | 80.0 | 77.4 | 77.4 | 78.8 |
| PhonemeSpectra | Best | 56.3 | 57.1 | 59.1 | 57.6 | 60.3 | 58.9 |
|  | Avg | 38.2 | 38.7 | 46.5 | 42.9 | 44.5 | 47.7 |
| LSST | Best | 78.7 | 79.5 | 79.9 | 74.6 | 75.7 | 79.9 |
|  | Avg | 69.3 | 69.4 | 71.2 | 61.4 | 61.4 | 67.3 |
| UWaveGestureLibrary | Best | 92.7 | 88.5 | 87.5 | 94.8 | 99.0 | 94.8 |
|  | Avg | 76.8 | 79.9 | 83.4 | 81.3 | 83.5 | 85.4 |
| CharacterTrajectories | Best | 99.0 | 98.0 | 99.6 | 98.7 | 99.3 | 98.4 |
|  | Avg | 93.6 | 94.2 | 96.6 | 96.6 | 95.2 | 97.0 |
| AsphaltPavementTypeCoordinates | Best | 72.3 | 77.7 | 81.1 | 89.5 | 88.5 | 89.5 |
|  | Avg | 72.7 | 75.8 | 77.0 | 81.2 | 82.1 | 83.7 |
| MotorImagery | Best | 87.5 | 83.3 | 77.1 | 79.2 | 66.7 | 87.5 |
|  | Avg | 84.9 | 83.3 | 83.2 | 73.6 | 74.0 | 81.9 |

## A.7 Attention Maps

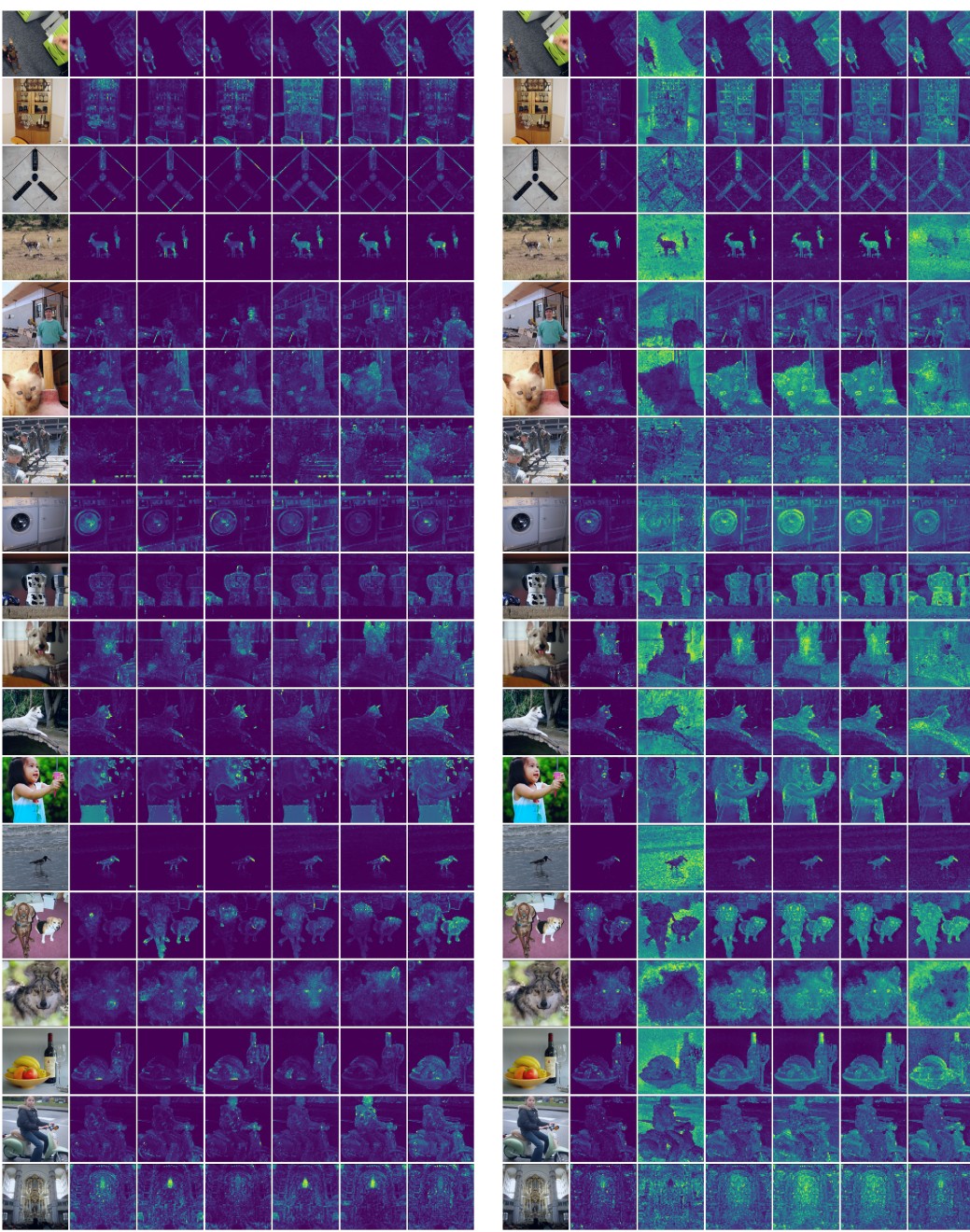

(a) Attention maps of the Jumbo token split into 6 smaller global tokens. Like ViT+Registers, ViT+Jumbo learns relatively artifact-free attention maps (as compared with the attention maps in (Darcet et al., 2024)).

(b) Attention maps of the CLS and the first five register tokens.

