# OpenReview forum: "Thicker and Quicker: The Jumbo Token for Fast Plain Vision Transformers"
_ICLR.cc/2026/Conference — ICLR 2026 Poster_

### Official Review · Reviewer_j2Lh · 2025-10-30

**Soundness:** 2
**Presentation:** 3
**Contribution:** 1
**Rating:** 2
**Confidence:** 5

**Summary:**

This paper introduces Jumbo, a simple yet effective extension of Vision Transformers (ViTs) that replaces the conventional single CLS token with a wider one. The “Jumbo” token is divided before self-attention and then reassembled afterward, enabling more powerful global representation learning at minimal additional cost. The approach preserves the elegance and simplicity of standard ViTs while integrating seamlessly with techniques such as token dropping and multimodal learning. Extensive experiments demonstrate consistent performance gains across various benchmarks, including ImageNet-1K and ImageNet-21K.

**Strengths:**

The Jumbo design can be seamlessly integrated into existing ViT architectures without requiring substantial structural modifications. It consistently outperforms both ViT+Registers and several optimized models, with particularly strong gains observed in smaller model settings. The paper provides extensive empirical evidence, including comprehensive comparisons, ablation studies, and diverse use-case evaluations.

**Weaknesses:**

Compared with ViT+Registers, the primary difference lies in using a single large token instead of multiple smaller ones—a design choice that may appear incremental rather than fundamentally innovative. Although Jumbo achieves notable performance gains, it comes at the cost of increased computation and a larger number of parameters, which may limit its practicality in real-world applications. The contributions are incremental.

**Questions:**

The proposed approach can potentially have a big increase in the cost of efficiency with a little performance gain.

---

> ### Author Response · Authors · 2025-11-20
> **1/1**
>
> We thank the reviewer for their emphasis on the importance of architecture, efficiency, and performance, and we fully agree on their importance. We are happy to address concerns about the novelty of the Jumbo architecture and its computational efficiency vs. its performance.
>
> > may appear incremental rather than fundamentally innovative
>
> Please recall that Registers are a small but important difference compared to the original ViT, i.e., several global tokens vs. one global token. Registers improve performance on ImageNet-1K by 0.5%, on ADE20K by 0.2%, and reduce attention-map artifacts by shifting high-norm outliers to register tokens that are not used in classification. Registers was an ICLR 2024 Oral and Award paper and has already been cited ~700 times. Registers' importance and impact stem mainly from it being a general and simple upgrade to the original ViT, which is _widely_ used (e.g., ~80K citations).
>
> Our Jumbo replaces the CLS token in plain ViTs with our extra-wide Jumbo token, paired with a single extra-wide MLP to efficiently increase network capacity—introducing asymmetric-width to transformers—while remaining attention-only and non-hierarchical for practicality. Our claim is _not_ that Jumbo is a radically new architecture, but that it improves the computational efficiency of plain ViTs in a technically novel way. Specifically, Jumbo improves supervised learning over Registers by up to 13% on ImageNet-1K (Fig. 1), self-supervised learning by 4.9% on ImageNet-1K (measured by linear probing, Tab. 1), and test-time adaptation by 5.2% on ImageNet-C (Tab. 2). Jumbo also reduces attention-map artifacts and eliminates high-norm tokens (Fig. 5), which translates into ADE20K segmentation gains of +1.9% and +2.1% for ViT-Base and ViT-Small scales, respectively, _over_ Registers models.
>
> > it comes at the cost of increased computation and a larger number of parameters, which may limit its practicality in real-world applications
>
> We agree with the reviewer that efficiency matters and can facilitate real-world use. We thoroughly demonstrate that Jumbo _improves_ computational efficiency—e.g., 1.9x faster for the same accuracy on ImageNet-21K (Fig. 1) and >10% more accuracy for the same speed on ImageNet-1K, ViT-nano size (Fig. 4)—thus making it more useful than Registers in real-world applications. Jumbo’s increase in model parameters is modest due to sharing weights across layers. Model parameters are only costly through increasing memory consumption, which in our case is also modest, as we measure in our benchmarking. For example, Jumbo improves accuracy on ImageNet-21K by 3.1% at a small cost to memory (↑300 MB at a 512 batch size). Furthermore, in our analysis presented in Tab. 5, we equalize the number of parameters between our Jumbo (row #1) and Registers (row #3); these models differ in accuracy by 0.1%, yet Jumbo is far more efficient with 1.7× the throughput, 0.5× the FLOPs, and 0.9× the memory.

---

### Official Review · Reviewer_rHVB · 2025-11-01

**Soundness:** 2
**Presentation:** 2
**Contribution:** 3
**Rating:** 6
**Confidence:** 4

**Summary:**

The paper proposes ViT+Jumbo which is a attention-only and non-hierarchical design that increases global capacity by replacing the CLS token with a single Jumbo token. To increase the model capacity, the Jumbo token is wider than patch tokens and has its own FFN whose parameters are shared across layers. The Jumbo token is split before MHSA and re-concatenated after attention. Only the Jumbo token passes through the Jumbo-FFN, keeping overhead small. The authors provide extensive experiments results to verfr their modification design on ImageNet-1K/21K, MAE pretraining, test-time adaptation, and time-series classification. Experimental results suggest that ViT+Jumbo improves speed-accuracy trade-offs versus ViT baselines and competes with specialized efficient backbones while preserving ViT compatibility.

**Strengths:**

1. Good practicality: The method remains attention-only and non-hierarchical, then the Jumbo token simply widens global processing while keeping the ViT interface intact. This preserves applicability of token dropping, SSL, and ViT heads. The design requires only two splits and concats plus a shared FFN with minimizing implementation burden while adding capacity where it matters.
2. Increasing capacity while maintaining throughput is promising: On ImageNet-1K the method attains or approaches the frontier across speeds and outperforming register-base models especially at small sizes. This shows effectiveness under constrained regimes.
3. This paper tests the proposed scheme from different experimental perspectives and tasks (supervised classification, SSL, TTA, and time-series classification), and provides a wealth of ablation experimental results and a relatively detailed explanation of the effects of hyperparameter tuning, which is commendable.
4. Some visual comparison results are provided, especially the visualization of the norm range, which can be compared according to the assumption of its most relevant baseline (i.e., the Register method).

**Weaknesses:**

1. Fairness and measurement choices may bias comparisons: ImageNet-1K training uses DeiT-III distillation and a specific two-stage recipe. But it may unclear that all baselines were tuned equivalently under this recipe. Without such distillation training, could the proposed schedule works well to coverage?
2. No experiments for downstream tasks like the common compared detection and segmentation which are important for verifying the robustness of vision encoders. So downstream dense tasks remain unverified as vision processing is the main field of the work. Besides, how to utilize Jumbo tokens at different layers in different downstream tasks is also worth exploring.
3. In the MAE self-supervised learning experiment, the linear probing setting was used to illustrate that performance cannot fully reflect the model's capabilities. Full fine-tuning is masking SSL, which is a more common choice, including MAE. It is better to add such results.

**Questions:**

1. Why "While many non-ViT architectures are both fast and accurate, they cannot flexibly process other input shapes" in the abstract? Non-hierarchical non-ViT models such as Vim and V-RWKV are also sequence models. How do they differ from ViTs in terms of input shape?
2. How do register-based methods compare to Jumbo in terms of performance, given similar model capacities?
3. Why does the Jumbo method work more effectively on smaller models? Can it be understood that the benefits are more pronounced with larger model sizes?
4. Please refer to the section on Weaknesses for other questions.

---

> ### Author Response · Authors · 2025-11-20
> **1/3**
>
> We thank the reviewer for the thoughtful and specific feedback. Our response provides clarifications and new results as requested on dense prediction, fine-tuning, and layer-wise analysis.
>
> > But it may unclear that all baselines were tuned equivalently under this recipe. Without such distillation training, could the proposed schedule works well to coverage?
>
> This is a standard setting [Yun & Ro, 2024, Qin et al., 2025], although multiple standards exist. We chose this approach to make the most of our available computational resources. We confirm that all baselines were indeed tuned equivalently and present the tuning results in the appendix; we will emphasize this point in our updated manuscript. Furthermore, our ImageNet-21K experiments (Section 4.2) did _not_ leverage distillation; here, Jumbo also outperforms ViT+Registers, e.g., by 3.1% and 1.2% at ViT-Small and ViT-Base scales, respectively.
>
> Jumbo improves on Registers with or without distillation.
>
> > how to utilize Jumbo tokens at different layers in different downstream tasks
>
> We agree this is interesting, and run experiments on 5 different downstream tasks. We report the best accuracy and its layer on each dataset for both Jumbo and Registers models, at the ViT-Base scale, that were originally trained via supervised learning on ImageNet-1K:
>
> | Dataset | Jumbo Accuracy | Jumbo Layer | Registers Accuracy | Registers Layer |
> |---------|---------------|-------------|-------------------|-----------------|
> | EuroSat | 98.0% | 8 | 97.6% | 9 |
> | Pets | 95.0% | 12 | 94.3% | 12 |
> | Caltech101 | 93.4% | 11 | 93.4% | 12 |
> | sNORB-Azimuth | 18.8% | 9 | 18.3% | 8 |
> | Clever-Distance | 70.8% | 8 | 67.2% | 8 |
>
> Full details. We choose these datasets because they are (1) commonly used, (2) diverse, i.e., structured, specialized, and natural types, and (3) are small enough to run during this rebuttal period. For each dataset and backbone (Jumbo and Registers), we train a linear probe on each of the 12 layers separately. We tune the probes by sweeping learning rates to ensure we are correctly assessing the information stored at the layer.
>
> > Downstream dense tasks
>
> Thank you for the suggestion. During the rebuttal period, we ran new experiments to determine whether Jumbo helps with dense tasks. Our Jumbo achieves mIoU of 44.4% and 39.1%, while Registers achieves mIoU of 42.5% and 37.0% for ViT-Base and ViT-Small sizes, respectively, on the ADE20K segmentation benchmark [R1]. We chose this dataset because it is the most commonly used semantic segmentation benchmark. We chose segmentation because it is a more local/higher-dimensional task to assess the effect of the global Jumbo vs. Register tokens. More specifically, we fine-tune on ADE20K for 100 epochs with a batch size of 16 and sweep learning rates {1e-5, 2e-5, 4e-5, 8e-5} for each model size {ViT-Base, ViT-Small} and global strategies {Jumbo, Registers}. We initialize model parameters from our ImageNet-1K supervised training experiments. We also attach a Segmenter head [R2], a commonly used segmentation head for ViTs.
>
> > Full fine-tuning for models pre-trained with MAE
>
> We agree with the reviewer that full fine-tuning of our Jumbo MAE model would further demonstrate its transferability. We thus run experiments comparing our Jumbo MAE with MAE’s official checkpoint—both models use ViT-Base backbones, pre-trained for 1600 epochs. Our Jumbo MAE either matches or exceeds the baseline in all cases. Considering that Jumbo significantly improves linear probing by 4.9% (Tab. 1), Jumbo is a general upgrade to plain ViTs for transfer. Please see the following results (all results are top-1 accuracy):
>
> ImageNet-1K: both models achieve 78.9%
>
> EuroSat: 98.7% (Jumbo) vs. 98.6% (baseline)
>
> Pets: 92.5% (Jumbo) vs. 91.9% (baseline)
>
> Caltech101: 91.6% (Jumbo) vs. 91.6% (baseline)
>
> sNORB-Azimuth: 25.9% (Jumbo) vs. 25.3% (baseline)
>
> Clever-Distance: 76.8% (Jumbo) vs. 75.6% (baseline)
>
> Full details. We choose ImageNet-1K because it is the gold-standard classification benchmark, and the other datasets because they are (1) commonly used, (2) diverse, i.e., structured, specialized, and natural types, and (3) are small enough to run during this rebuttal period. For each dataset and backbone (Jumbo and the baseline), we sweep learning rates {1e-5, 3e-5, 1e-4, 3e-4} with a 128 batch size. On ImageNet-1K we fine-tune for 10 epochs, and on the other datasets, we fine-tune for 100 epochs (since they are smaller). We perform all experiments in a single codebase for fairness. We have started 100-epoch runs on ImageNet-1K and will post the results during this discussion period when they finish.

---

> ### Author Response · Authors · 2025-11-20
> **2/3**
>
> > Why "While many non-ViT architectures are both fast and accurate, they cannot flexibly process other input shapes" in the abstract? Vim and V-RWKV are also sequence models.
>
> As the review points out, both Vim  [R10] and V-RWKV [R11] are _not_ ViTs nor attention-only; thus, they require further customization to process other shapes. Plain ViTs simply require defining position embeddings to accommodate new shapes, which is trivial (e.g., one line of code). We will modify this sentence in our abstract to better reflect this point and thank the reviewer for their attention to detail.
>
> Plain ViTs (including with Registers or Jumbo) are flexible to input shapes:
>
> (1) ViTs can drop/mask tokens to process any subset of patches simply via indexing, which is not possible for other architectures. For example, SHViT, MobileNetV4, and EfficientViT all use convolutions and pooling. Mamba-based architectures, such as Vim, require defining a patch-order, e.g., left-to-right. Dropping patches in such a sequence model incorrectly assigns their positions, unlike transformers. Out of the box, V-RWKV’s “Q-Shift” does not support token dropping since it is convolutional, but the authors of V-RWKV modify it to enable MAE pre-training. Token dropping/masking is used in all SOTA SSL algorithms, can speed up training and inference (e.g., through adaptive computation), and is thus a desirable trait of model architectures.
>
> (2) ViTs can process non-2D data, for example, remote sensing inputs or combinations of _different_ shapes. SOTA remote sensing foundation models process inputs of shape: height X width X spectral groups X time [R3, R4]. Applying Mamba-based architectures to this input would require defining complex (and unnatural) scans over each axis. Applying V-RWKV to this input would require re-designing the Q-Shift operation. Plain ViTs require defining position embeddings, which is trivial and explains their dominance in remote sensing and other applications. Plain ViTs can also process inputs composed of different shapes, e.g., a joint language-vision encoder, via position embeddings. SHViT, MobileNetV4, EfficientViT, Vim, and V-RWKV would all require significant customization to _jointly_ encode 1D (language) and 2D (vision) inputs.
>
> Furthermore, our Jumbo token and shared MLP can be incorporated into the V-RWKV architecture, which we save for future work. We thank the reviewer for the question and we will revise our manuscript, specifically section 2.2, to discuss V-RWKV and Vim.
>
> > How do register-based methods compare to Jumbo in terms of performance, given similar model capacities?
>
> We agree that comparing Jumbo with Registers for the same model capacity is an important question. When we focus on the number of model parameters as a way to quantify “model capacity”, in Tab. 5, we can compare our Jumbo (row #1) with a Registers model with _equal_ parameters (row #3). These two models differ in accuracy by 0.1%, yet Jumbo is _far_ more efficient with 1.7× the throughput, 0.5× the FLOPs, and 0.9× the memory.
>
> We note that model parameters contribute to model capacity, but it is not the only factor to consider. For example, models can share weights across many layers to achieve more capacity per parameter or increase the input size, which shares parameters across the feature map / tokens. Hence, we report several measures of computation (parameters, FLOPs, throughput, and memory), and we focus on throughput since it best reflects a model’s true cost per image on current hardware.

---

> ### Author Response · Authors · 2025-11-20
> **3/3**
>
> > Why does the Jumbo method work more effectively on smaller models?
>
> Prior ViTs decrease the embedding dimension of all tokens (patch tokens and global tokens) to increase speed. Our Jumbo decreases the embedding dimension of patch tokens for speed _and_ retains global capacity via our extra-wide global token for accuracy. Our results indicate that Jumbo’s gains over other ViTs are largest when models are faster/smaller and their outputs are larger, which we attribute to a greater need for global capacity, which is increased by the Jumbo token. We show this by training ViTs of many sizes, where Jumbo gains are largest (e.g., 13%+) at the ViT-Nano scale.
>
> Furthermore, we show that increasing the output dimensionality results in more gain from Jumbo. We increase the output size in terms of the number of classes via training on _ImageNet-21K_, on which Jumbo outperforms ViT+Registers, e.g., by 3.1% and 1.2% at ViT-Small and ViT-Base scales (Fig 1b. and Sec. 4.2). As a rebuttal experiment, we also increase the output size in terms of spatial resolution by pixel-wise prediction (i.e. semantic segmentation) on ADE20K, which has 150 classes and 512x512 spatial output, where Jumbo achieves mIoU of 44.4% and 39.1% while Registers achieves mIoU of 42.5% and 37.0% for ViT-Base and ViT-Small sizes respectively. Here, we explain Jumbo’s performance by its higher global capacity for modeling spatial context and its elimination of high-norm tokens (Fig. 5), which result in patch-token representations that are more faithful to their original position [Darcet et al., 2024] as needed for the spatial positions of pixel-wise prediction. For more details on ADE20K please see "Downstream dense tasks" in the response to Reviewer rHVB.
>
> [R1] Scene parsing through ade20k dataset, Zhou et al., CVPR 2017
>
> [R2] Segmenter: Transformer for Semantic Segmentation, Strudel et al., ICCV 2021
>
> [R3] Olmo-Earth: https://allenai.org/olmoearth
>
> [R4] Galileo: Learning Global & Local Features of Many Remote Sensing Modalities, Tseng and Fuller et al., ICML 2025
>
> [R5] EuroSAT: A Novel Dataset and Deep Learning Benchmark for Land Use and Land Cover Classification, Helber et al. IEEE Journal of Selected Topics in Applied Earth Observations and Remote Sensing 2019
>
> [R6] Cats and dogs, Parkh et al., CVPR 2012
>
> [R7] One-shot learning of object categories, Li et al., TPAMI 2006
>
> [R8] Learning methods for generic object recognition with invariance to pose and Lighting, LeCun et al., CVPR 2004
>
> [R9]  Clevr: A diagnostic dataset for compositional language and elementary visual reasoning, Johnson et al., CVPR 2017
>
> [R10] Vision Mamba: Efficient Visual Representation Learning with Bidirectional State Space Model, Zhu and Liao et al., ICML 2024
>
> [R11] Vision-RWKV: Efficient and Scalable Visual Perception with RWKV-Like Architectures, Duan and Wang and Chen et al., ICLR 2025

---

> ### Author Response · Authors · 2025-11-26
> **update on our first Jumbo MAE 100-epoch fine-tuning**
>
> Our 100-epoch fine-tuning runs on ImageNet-1K have finished. Our _first_ Jumbo MAE achieves 82.6% accuracy, and the baseline MAE achieves 83.3% accuracy. This is an encouraging result considering that we use MAE’s hyper-parameters without tuning. Should our paper be accepted we will perform a small hyper-parameter search—under the 100-epoch fine-tuning regime—to ensure a fair comparison with the baseline MAE, which has been further tuned.
>
> Please recall that with _equal_ tuning, our Jumbo MAE outperforms or ties the baseline MAE under fine-tuning (please see our response to “Full fine-tuning for models pre-trained with MAE”). And our Jumbo MAE outperforms the baseline MAE by 4.9% under linear probing (Tab. 1).

---

> > ### Comment · Reviewer_rHVB · 2025-11-26
> >
> > Thanks to the authors for their detailed replies and most of my concerns have been addressed. I tend to be positive about accepting this work and maintain the positive rating.

---

### Official Review · Reviewer_oXPN · 2025-11-02

**Soundness:** 3
**Presentation:** 3
**Contribution:** 3
**Rating:** 8
**Confidence:** 3

**Summary:**

This paper proposes adding a jumbo token to ViTs, demonstrating good efficiency compared to ViT register baselines. The paper is well-written and the idea is intuitive.

**Strengths:**

1. The paper is well-written.
2. The authors perform systematic and high-quality ablation studies.

**Weaknesses:**

1. Thanks for the authors' discussions in Sec. 4.7, which show that the additional parameters indeed improve the accuracy. The advantage of Jumbo token is the efficiency, which limits the upper bound of this method.

2. The current use case seems limited. I am curious to see its potential in DiT for C2I or T2I image generation.

3. As discussed in the limitation section, does this method have potential in visual-language understanding or language models?

4. Can the authors explain and analyze why Jumbo reduces the high norms of tokens more effectively than Register as shown in Fig. 5?

**Questions:**

1. Can the authors report the model parameters in Tab. 5 for the Alt.1 and Alt. 2 model variants?
2. Why did the authors perform ablations on different scales of ViTs in Tab. 5? Would the conclusions change for Alt. 1 and Alt. 2 if using ViT-Base models?

---

> ### Author Response · Authors · 2025-11-20
> **1/2**
>
> We thank the reviewer for the thoughtful and specific feedback. Our response provides clarifications and new results as requested on language tasks and alternate architectures.
>
> > The advantage of Jumbo token is the efficiency, which limits the upper bound of this method.
>
> With supervised training on ImageNet-1K, the gap between Jumbo and Registers indeed narrows with increasing model size—effectively closing at the ViT-Base scale. However, at this same model size of _ViT-Base_, Jumbo outperforms Registers by 1.2% on _ImageNet-21K_ (Fig. 1), by 4.9% on ImageNet-1K linear probing after MAE pre-training (Tab. 1), and by 1.9% on ADE20K segmentation (as provided in a rebuttal experiment in the response to Reviewer rHVB). This shows that Jumbo can provide advantages across a range of tasks and model sizes.
>
> > potential in visual-language understanding or language models?
>
> We thank the reviewer for this suggestion. During this rebuttal period, we ran two experiments to investigate Jumbo’s potential for visual-language understanding and masked language modeling. Jumbo helps in both settings.
>
> **Jumbo can help with visual-language understanding.** We aligned our Jumbo and Register models (ViT-Base scale) to language embeddings. Jumbo significantly outperforms Registers in these new experiments. Specifically, Jumbo gains +4% over Registers for a caption retrieval task, and +5.5% for zero-shot classification on ImageNet-1K.
>
> Full details. We fine-tune our Jumbo and Register models (ViT-Base scale), that we originally trained on ImageNet-1K, for language alignment. We chose the CC3M dataset [R3], a standard image-caption dataset containing 3 million pairs. We then created caption embeddings using the Qwen3 embedding model, a SOTA text embedding model. We fine-tuned to minimize the mean squared error between caption-embedding predictions made by the ViT given images, and Qwen3’s embeddings. We optimized for 500K steps with a batch size of 128, taking ~2 days per run on a 4090 GPU. We chose this procedure because we do not have the computational resources to train a CLIP-like model, which requires several thousand samples per batch to generate negatives; we thus opt for language-alignment _fine-tuning_ with a SOTA frozen text embedder. After fine-tuning, our Jumbo achieved a loss of 4.3e-4 on CC3M’s validation set, compared to Register’s 4.4e-4. This loss gap is subtle but translates to significant accuracy gains for Jumbo. To check accuracy on an image-caption retrieval task, we embed all captions in CC3M’s validation set with Qwen3 and embed all corresponding images with our fine-tuned ViTs. Jumbo’s image embeddings retrieved the correct caption embedding 34% of the time in the top 10, whereas Registers scored 30%. To assess language alignment by other means, we embed all ImageNet-1K class names with Qwen3 (only once per class) and ImageNet-Val images with our fine-tuned vision encoders. We then computed the closest class embedding to each image embedding to assign the image a class prediction (equivalent to kNN with k=1). Jumbo achieved a top-1 accuracy of 23.5% and Registers achieved 18%. These comparisons are fair because both models used the same pre-training and fine-tuning recipes, datasets, and computational budgets.
>
> **Jumbo can help with masked language modeling.** We trained Jumbo and Registers (base sizes) using BERT’s language modeling objective. Jumbo achieved a validation perplexity of 4.8, while Registers achieved 4.9 (lower is better). MLM perplexity correlates with performance on downstream tasks.
>
> Full details. We train Jumbo and Register models (base size) from scratch using BERT’s language modeling objective and hyper-parameters. We chose the C4 dataset [R4], which is a commonly used pre-training dataset for models of this scale. We optimized for 500K steps with a batch size of 128, taking ~1.5 days on a 4090 GPU. We report perplexity on C4’s validation set. We chose to only report perplexity, and not performance on many downstream tasks, because perplexity correlates with downstream performance [R2], and fine-tuning these models on several datasets cannot be achieved in the allotted time.
>
> We emphasize that these language-alignment and masked language modeling experiments are meant to show _potential_ in these areas—they are not meant to achieve SOTA results. We hope the promise that Jumbo shows in vision-language and language-only tasks will inspire NLP researchers to leverage Jumbo. If accepted, we will have an extra page to include these positive proof-of-concept results in a new subsection between 4.5 and 4.6, as further evidence of the generality of Jumbo.

---

> ### Author Response · Authors · 2025-11-20
> **2/2**
>
> > I am curious to see its potential in DiT for C2I or T2I image generation.
>
> We appreciate the reviewer’s interest in applying Jumbo to image generation. However, our paper is focused on recognition/understanding, not generation; we will update our abstract to clarify our scope. Furthermore, our lab does not have the expertise in diffusion modeling nor sufficient computational resources to carefully perform image-generation experiments during the allotted time. Because Jumbo eliminates high-norm tokens that distort position-specific representations, Jumbo may indeed help image generation, as it does other dense tasks, e.g., semantic segmentation, and we certainly encourage others to pursue this line of research and we will release our code and models on publication.
>
> > why Jumbo reduces the high norms of tokens more effectively than registers
>
> We offered Fig. 5 as an empirical result, as readers may be curious if Jumbo also reduces high norms. Darcet et al. claim that high norms represent global information within tokens, and that registers provide tokens for ViTs to store this information elsewhere, thereby keeping patch-token representations faithful to their positions (reducing artifacts). In Fig. 5, our ViT+Jumbo (J=6) has fewer global tokens than ViT-Registers (16 registers), yet our Jumbo has fewer high norms (zero in fact). This finding demonstrates that Jumbo's single extra-wide MLP effectively eliminates high norms. This conclusion may be related to Jiang and Dravid et al.'s [R1] recent finding that high norms in ViTs originate in the MLPs of patch tokens. We will include this discussion in Section 4.7 of our updated manuscript.
>
> > model parameters in Tab. 5 for the Alt.1 and Alt. 2 model variants
>
> Tab. 5 shows the parameter count of all models in column 3 (“params”). Alt. 1 and Alt. 2 models have 40M parameters.
>
> > Why did the authors perform ablations on different scales of ViTs in Tab. 5?
>
> Most of our ablation and analysis experiments were performed at the ViT-Small scale to manage computational cost. We conducted an _extra_ experiment at the ViT-Base scale, Tab. 5, to match parameters between Jumbo (row #1) and Registers (row #3) by widening _all tokens_ in a ViT+Registers model. This demonstrates that **asymmetric token-widening**—which we introduce—is superior to widening all tokens (i.e., the standard way of increasing ViT width). Specifically, Jumbo maintains accuracy while being far more efficient with 1.7× the throughput, 0.5× the FLOPs, and 0.9× the memory compared to a ViT+Registers variant with equal parameters.
>
> > Would the conclusions change for Alt. 1 and Alt. 2 if using ViT-Base models?
>
> We also expect Jumbo to outperform these alternative methods at the ViT-Base scale. To check, we started running the requested experiments, which should complete by Nov. 24 during this discussion period (each ImageNet-21K run at the ViT-Base scale takes ~7 days). Thus far, at epoch 33/50, alt. #1 and alt. #2 both achieve 38.9%, while our Jumbo achieved 40.1%. This follows the trend from ViT-Small. We will post again when these experiments are completed.
>
> [R1] Vision Transformers Don’t Need Trained Registers, Jiang and Dravid et al., NeurIPS 2025
>
> [R2] Scaling Laws for Transfer, Hernandez et al., arXiv 2021
>
> [R3] Conceptual 12M: Pushing Web-Scale Image-Text Pre-Training To Recognize Long-Tail Visual Concepts, Changpinyo et al., CVPR 2021
>
> [R4] Exploring the limits of transfer learning with a unified text-to-text transformer, Raffel et al., JMLR 2020

---

> ### Author Response · Authors · 2025-11-26
> **update on alternative architectures**
>
> The requested ImageNet-21K experiments using alternative architectures at ViT-B scale have completed: Alt. #1 achieves 46.0% and Alt. #2 achieves 46.4%. Our Jumbo outperforms the better of these two alternatives by 0.6% in top-1 accuracy and is 11% faster while matching parameter counts and memory usage. This confirms the improvement of Jumbo vs. alternate FFN designs alongside our existing results at ViT-S scale and we will include both in the revision.

---

### Author Response · Authors · 2025-12-04
**Summary of Discussion**

To the AC,

We would like to summarize the discussion to help inform the decision. Reviewers agree that our Jumbo is general and maintains the ViT interface (rHVB, j2Lh), that Jumbo is efficient and improves on ViTs + Registers (oXPN, j2Lh), and the experiments thoroughly ablate its design (oXPN, rHVB, j2Lh) and show multiple use cases (j2Lh, rHVB).

- Reviewer oXPN was concerned that Jumbo’s upper bound on accuracy is limited. Jumbo achieves _large_ gains at the smallest/fastest scales (e.g., up to 13% on ImageNet-1K over ViT+Registers at ViT-Nano scale, Figure 1). We demonstrated that Jumbo achieves strong gains at the ViT-Base scale as well, e.g., on ImageNet-21K (+1.2%, Table 5), linear probing after MAE pre-training (+4.9%, Table 1), and ADE20K segmentation (+1.9%, rebuttal).
- Reviewer oXPN asked for language-image or language-only experiments. During the rebuttal, we demonstrated positive proof-of-concept results for both settings (+4% on image-caption retrieval and lower perplexity in masked language modeling).
- Reviewer oXPN liked our ablations and asked for another experiment using alternative architectures at a different scale (ViT-B, rather than ViT-S). We ran this experiment, confirming that our Jumbo’s asymmetric width explains its effectiveness.
- Reviewer rHVB asked for dense prediction experiments. During the rebuttal, we ran new experiments on ADE20K for semantic segmentation, which demonstrate that Jumbo outperforms ViT+Registers by significant margins (+2%) under an equal hyperparameter tuning budget.
- Reviewer rHVB asked for experiments that fine-tune our Jumbo that was pre-trained via masked autoencoding (MAE). During the rebuttal, we ran new experiments on ImageNet-1K and 5 other common downstream image classification tasks, our Jumbo either tied or beat the baseline under an equal hyperparameter tuning budget.
- Reviewer rHVB asked if Jumbo outperforms ViT+Registers with similar capacity. In our submission, we thoroughly demonstrated that our Jumbo outperforms ViT+Registers with similar capacity, where capacity is measured by throughput (Figures 1, 4), FLOPs (Table 5), and parameters (Table 5: "Registers D: 768→1024").
- Reviewer rHVB asked for clarification on why architectures like Vision Mamba (Vim) or V-RWKV cannot flexibly handle different input shapes (e.g., token masking or vision-language modalities). We responded with a detailed discussion: briefly, ViT’s attention-only and non-hierarchical design enables trivial flexibility over input shapes—which our Jumbo _maintains_ for generality.
- Reviewer j2Lh had two concerns: novelty and computational efficiency. On efficiency, our submission thoroughly demonstrates that our Jumbo is more computationally efficient than ViT+Registers, and also highly specialized compute-efficient architectures like EfficientViT and SHViT (Figure 1, Figure 4). Regarding novelty, we reminded the reviewer that Registers are a small but important difference compared to the original ViT. Registers was an ICLR 2024 Oral and Award paper and has already been cited ~700 times. Jumbo introduces an extra-wide global token resulting in asymmetric token width (a first for transformers), while maintaining plain ViT compatibility for practicality. Finally, Jumbo’s gains over ViT+Registers are much greater than ViT+Registers gains over the original ViT (e.g., +0.5% on IN-1K and +0.2% on ADE20K)—and Jumbo eliminates the high-norm tokens which Registers only reduce (Figure 5).

We sincerely thank the AC for their service to the community.

---

### Meta-Review · Area_Chair_C6xu · 2026-01-07

**Summary:**

This paper proposes Jumbo, a simple yet effective architectural modification to plain Vision Transformers that improves efficiency by asymmetrically increasing global capacity. Specifically, the method replaces the standard CLS token with a single extra-wide Jumbo token processed by its own FFN, while keeping patch tokens narrow. The design preserves the attention-only, non-hierarchical ViT interface and introduces minimal computational overhead through parameter sharing. Extensive experiments across ImageNet-1K/21K, masked autoencoding, test-time adaptation, dense prediction, and time-series modeling demonstrate consistent improvements in speed–accuracy trade-offs over ViT and ViT+Registers baselines, while remaining competitive with specialized efficient architectures.

Reviewers were generally positive and raised questions mainly about `the degree of novelty relative to existing approaches` such as ViT+Registers, `the effectiveness of the method at larger model scales`, `the fairness of training protocols and baseline comparisons`, and `the generality of the approach to downstream tasks beyond image classification`. These concerns were mostly addressed through additional experiments, ablations, and clarifications in the rebuttal, which demonstrated that the proposed asymmetric token-width design yields consistent and non-trivial gains across tasks and settings.

The paper presents a clear, well-motivated architectural insight that is simple, novel, and broadly applicable: allocating model capacity asymmetrically to global tokens yields better efficiency without sacrificing the defining advantages of plain ViTs. The method is easy to adopt, well supported by thorough experiments and ablations, and shows consistent benefits across tasks and modalities. The rebuttal convincingly addressed reviewer concerns and further strengthened the empirical case. Overall, this work represents a meaningful and practical contribution and is recommended for acceptance.

**Reviewer Concerns:**

**1. [Addressed] Upper-bound accuracy and effectiveness at larger model scales
(by: `oXPN`)**

Reviewer `oXPN` questioned whether the proposed Jumbo token mainly improves efficiency at small model scales and whether its accuracy gains diminish as model size increases, potentially limiting its usefulness for larger ViTs.

The rebuttal addressed this concern by providing additional results at the ViT-Base scale, including ImageNet-21K supervised training, MAE pre-training with linear probing, and ADE20K semantic segmentation. These results demonstrate that Jumbo continues to provide consistent gains over ViT+Registers at larger scales, alleviating concerns about a limited upper bound.

**2. [Addressed] Generality to downstream tasks beyond image classification
(by: `rHVB`, `oXPN`)**

Reviewers asked whether Jumbo generalizes to downstream tasks beyond ImageNet classification, particularly dense prediction tasks and other application domains.

In response, the authors added new experiments on ADE20K semantic segmentation, multiple downstream classification datasets, test-time adaptation, and time-series modeling, showing consistent improvements over ViT and ViT+Registers under comparable settings. These additions satisfactorily addressed concerns about downstream generality.

**3. [Addressed] Fairness of comparisons and training protocol choices**
(by: `rHVB`)

Reviewer `rHVB` raised concerns that the use of DeiT-III distillation and specific training recipes could bias comparisons against baselines, and questioned whether all models were equivalently tuned.

The authors clarified that all baselines were trained and tuned under equivalent protocols, provided additional details and results in the appendix, and further demonstrated Jumbo’s advantages in non-distillation settings (e.g., ImageNet-21K), resolving this concern.

**4. [Partially addressed] Applicability to multimodal and language-only settings
(by: `oXPN`)**

Reviewer `oXPN` asked whether the Jumbo design could extend to vision–language understanding or language-only models, noting that the paper primarily focuses on vision tasks.

The rebuttal added proof-of-concept experiments on image–caption retrieval and masked language modeling, showing positive results. While these experiments demonstrate promising generality, reviewers acknowledged that the evaluation remains preliminary, leaving this concern partially addressed.

**5. [Addressed] Architectural clarity and mechanism explanation
(by: `oXPN`, `rHVB`)**

Reviewers requested clearer explanations of why Jumbo reduces high-norm tokens more effectively than Registers and how the asymmetric token width contributes to improved performance.

The authors provided additional analysis and discussion linking the behavior of high-norm tokens to MLP capacity and global information storage, and clarified the architectural design with further ablations and visualizations. This concern was adequately addressed.

**6. [Partially addressed] Novelty relative to ViT+Registers
(by: `j2Lh`)**

Reviewer `j2Lh` questioned whether replacing multiple Register tokens with a single wider Jumbo token constitutes a sufficiently novel contribution, characterizing the change as incremental rather than fundamentally innovative.

The rebuttal clarified the conceptual distinction of introducing asymmetric token width while preserving a plain, attention-only ViT, and provided evidence that Jumbo achieves substantially larger gains over Registers than Registers achieve over the original ViT. While these clarifications and results strengthen the case for novelty, the reviewer remained unconvinced, leaving this concern partially addressed.

**7. [Addressed] Computational efficiency and parameter overhead
(by: `j2Lh`)**

Reviewer `j2Lh` expressed concern that the improved accuracy may come at the cost of increased computation and parameters.

The authors demonstrated that Jumbo achieves better accuracy–efficiency trade-offs than ViT+Registers and several specialized efficient architectures under matched throughput, FLOPs, and parameter settings. This concern was addressed.

**Reviewer Scores:**

**Reviewer oXPN (8 -> 8)**

Reviewer oXPN was positive throughout, highlighting the clarity of the idea, strong ablations, and consistent efficiency gains. The rebuttal addressed all raised concerns with additional experiments and analysis, reinforcing the original assessment.

**Reviewer rHVB (6 -> 7)**

Reviewer rHVB viewed the paper as marginally above the acceptance threshold, with concerns about fairness, downstream tasks, and SSL evaluation. These were comprehensively addressed in the rebuttal, and the reviewer explicitly indicated increased confidence and a positive stance during discussion, supporting a modest upward revision.

**Reviewer j2Lh (2 -> 3)**

Reviewer j2Lh maintained a strong negative view, primarily due to a disagreement on the level of novelty relative to ViT+Registers. While the rebuttal provided detailed clarification and empirical evidence addressing efficiency and architectural distinctions, this did not change the reviewer’s fundamental assessment. From an AC perspective, the score would likely have a minor change.

From an AC perspective, although this concern reflects a reasonable difference in judgment regarding novelty, it is outweighed by the clarity of the architectural insight, the simplicity and ease of adoption of the method, and the strength and breadth of empirical validation. The contribution is viewed as a meaningful and practical architectural improvement rather than a minor tweak, and thus the paper is considered worthy of acceptance despite this remaining disagreement.

---

### Decision · Program_Chairs · 2026-01-26

Accept (Poster)